

# Identification and characterization of the CRK gene family in the wheat genome and analysis of their expression profile in response to high temperature-induced male sterility

Hongzhan Liu[1,2,3], Xiaoyi Li[1], Zehui Yin[1], Junmin Hu[4], Liuyong Xie[1], Huanhuan Wu[1], Shuying Han[1], Bing Li[1], Huifang Zhang[1], Chaoqiong Li[1], Lili Li[1,2,3], Fuli Zhang[1,2] and Guangxuan Tan[1,3]

[1] College of Life Science and Agronomy, Zhoukou Normal University, Zhoukou, Henan Province, China
[2] Field Observation and Research Station of Green Agriculture in Dancheng County, Dancheng, Henan Province, China
[3] Engineering Technology Research Center of Crop Molecular Breeding and Cultivation in Henan Province, Zhoukou, Henan Province, China
[4] Jiaozuo Seed Management Station, Jiaozuo, Henan Province, China

Corresponding authors
Hongzhan Liu,
liuhongzhan0111@sina.com
Guangxuan Tan, gxtan@zknu.edu.cn

## ABSTRACT

Cysteine-rich receptor-like kinases (CRKs) play many important roles during plant development, including defense responses under both biotic and abiotic stress, reactive oxygen species (ROS) homeostasis, callose deposition and programmed cell death (PCD). However, there are few studies on the involvement of the CRK family in male sterility due to heat stress in wheat (*Triticum aestivum* L.). In this study, a genome-wide characterization of the CRK family was performed to investigate the structural and functional attributes of the wheat CRKs in anther sterility caused by heat stress. A total of 95 CRK genes were unevenly distributed on 18 chromosomes, with the most genes distributed on chromosome 2B. Paralogous homologous genes with Ka/Ks ratios less than 1 may have undergone strong purifying selection during evolution and are more functionally conserved. The collinearity analysis results of CRK genes showed that wheat and *Arabidopsis* (*A. thaliana*), foxtail millet, *Brachypodium distachyon* (*B. distachyon*), and rice have three, 12, 15, and 11 pairs of orthologous genes, respectively. In addition, the results of the network interactions of genes and miRNAs showed that five miRNAs were in the hub of the interactions map, namely tae-miR9657b-5p, tae-miR9780, tae-miR9676-5p, tae-miR164, and tae-miR531. Furthermore, qRT-PCR validation of the six *TaCRK* genes showed that they play key roles in the development of the mononuclear stage anthers, as all six genes were expressed at highly significant levels in heat-stressed male sterile mononuclear stage anthers compared to normal anthers. We hypothesized that the *TaCRK* gene is significant in the process of high-temperature-induced sterility in wheat based on the combination of anther phenotypes, paraffin sections, and qRT-PCR data. These results improve our understanding of their relationship.

## INTRODUCTION

In living organisms, protein kinases play an important role in receiving signals, sensing them, and delivering them to effector genes. In plants, receptor-like protein kinases (RLKs) as conserved components of the signaling pathway, are a class of single transmembrane proteins localized to the cell membrane; they consist of many different protein species (*Sakamoto et al., 2012*). The extracellular structural domains of plant RLKs can recognize external stimuli such as extracellular growth, developmental and environmental factors. They are able to transmit this information to the intracellular kinase region for phosphorylation or dephosphorylation, thereby turning on or off downstream target proteins and thus regulating plant growth, development and various responses to adversity (*Morris & Walker, 2003*; *Wang et al., 2008*). RLKs can be classified into several subfamilies based on differences in the amino acid sequences of their extracellular structural domains, including domain of unknown function 26 (DUF26) RLKs, cysteine-rich repeat RLKs (CRKs), leucine-rich repeat RLKs (LRR-RLKs), lysine motif (LysM), S-domain RLKs, and others (*Shiu & Bleecker, 2001*; *Shiu et al., 2004*).

CRK is a class of membrane receptor proteins widely found in animals and plants, consisting of an N-terminal signal peptide, a transmembrane structural domain, a C-terminal serine/threonine protein kinase domain, and an extracellular domain containing one to four copies of the strss-antifung domain (*Chen, 2001*). Currently, 44 CRKs have been identified in *A. thaliana*. (*Chen, 2001*; *Burdiak et al., 2015*), 70 in cotton (*Zhang et al., 2018*), 35 in tomato (*Liu et al., 2021a*), 45 in rice (*Li et al., 2018*), and 46 in *Phaseolus vulgaris* (*Quezada et al., 2019*). CRKs play a significant role in plant growth and development, hormone signaling, abiotic stress and pathogen defense, and programmed cell death (PCD). In *A. thaliana*, *AtCRK5* and *AtCRK13* were rapidly induced by *Pseudomonas syringae*, and overexpression of *AtCRK5* or *AtCRK13* induced programmed cell death and rapid expression of pathogenesis-related proteins genes in the plants, thereby enhancing resistance to *Pseudomonas syringae* (*Chen, Du & Chen, 2003*; *Acharya et al., 2007*). Meanwhile, the CRK5 promoter region was found to contain a large number of WRKY transcription factor binding sites, and CRK5 may regulate *A. thaliana* senescence through WRKY53 and WRKY70 transcription factors (*Burdiak et al., 2015*). Plants that are infested by pathogenic bacteria transmit external signals into the cells to initiate the appropriate defense response. The immune response triggered by pathogen-associated molecular patterns (PAMP- triggered immunity) is primary immune defense through the production of reactive oxygen species (ROS) or the accumulation of callus; secondary immune defense in plants is generally due to the accumulation of salicylic acid (SA), which causes PCD (*Chisholm et al., 2006*).

In *A. thaliana*, the overexpression of *AtCRK4, AtCRK5, AtCRK13, AtCRK19,* and *AtCRK20* may lead to PCD (*Chen, Du & Chen, 2003*; *Chen et al., 2004*; *Acharya et al.,*

*2007*). *AtCRK4, AtCRK19,* and *AtCRK20* are significantly induced by pathogen infection and salicylic acid (*Chen et al., 2004*; *Ederli et al., 2011*). The overexpression of *AtCRK4, AtCRK6, AtCRK7,* or *AtCRK36* were enhanced the early and mid-term immune responses of plants against pathogenic bacteria (*Yeh et al., 2015*; *Lee et al., 2017*). Additionally, *AtCRK6* and *AtCRK7* are involved in mediating the production of extracellular ROS in *A. thaliana* (*Idänheimo et al., 2014*). The overexpression of *AtCRK28* or *AtCRK29* accelerated the outbreak and accumulation of extracellular ROS, which induced cell death in *A. thaliana*. Further, *AtCRK28* also affected flowering and development of *A. thaliana* (*Yadeta et al., 2016*).

The expression of tomato *SlCRK1* was also found to be significantly higher in the flowers than in the roots, leaves, fruits, and seeds, with some tissue-specific expression. The significant expression of *SlCRK1* in floral organs was also characterized by pollen-specific expression (*Kim et al., 2014*). Most *SlCRKs* downregulate their expression levels in response to heat stress and GO-enriched co-expressed genes possess similar expression patterns in response to disease and heat stressors (*Liu et al., 2021a*).

In previous studies, we found that the onset of male sterility can be caused by the application of appropriate heat stress to the developmental stages of pistil and stamen differentiation (*Liu et al., 2018a*; *Liu et al., 2021b*). The wheat genome is complex and contains many nutrients that are important for human health but wheat yield can be directly affected by various stresses during the plant's growth and development or by the occurrence of anther abortion due to heat stress during flowering (*Ullah et al., 2022*). Natural plants can sense external stresses in certain ways, and CRKs are involved in abiotic stress responses such as hormones, salt stress, heat stress, drought, and redox stress (*Nemhauser, Hong & Chory, 2006*; *Tanaka et al., 2012*). However, the regulatory network of the wheat anther response to heat stress is largely obscure, and the putative role of CRK genes in abiotic stress adaptation during wheat anther development, especially heat stress, has not been investigated.

We performed evolutionary tree construction, motif analysis, analysis of paralogous and orthologous homologous genes, and gene expression profiling from the perspective of the wheat *TaCRK* gene family to reveal the potential role of *TaCRK* genes in the heat stress response of wheat anthers. Here, we identified genome-wide members of the wheat *CRK* gene family and systematically analyzed the isoelectric point, molecular weight, subcellular localization, conserved motifs, cis-acting elements, and gene replication and gene expression patterns of all family members. A total of 95 putative *TaCRK* genes were identified and named according to their chromosomal distributions. In addition, the expression patterns of six *TaCRK* genes in wheat anther sterility under heat stress were examined by qRT-PCR. These results provide a theoretical basis and technical reference to better understand the functional role of the *CRK* gene family in male sterility in wheat subjected to stress.

## MATERIALS & METHODS

### Wheat material, paraffin sections and characterization of phenotypes

The wheat seeds of cv. Zhoumai 36 were sown on October 28, 2022, in the experimental field of Zhoukou Normal University, Zhoukou, Henan Province, People's Republic of China (33°C64′N, 114°C6′E). The wheat material was covered with plastic film in early April of the following year and all treatment procedures were carried out as described in our previously research (*Liu et al., 2018a*; *Liu et al., 2022*). Samples from normal anthers and HT-ms anthers were collected in six centrifuge tubes (1.5 mL), quick-frozen in liquid nitrogen and quickly stored at −80 °C. Three of the samples were used for qRT-PCR experiments within half a month and the remaining three were kept in reserve. In addition, anthers were treated with FAA fixative and placed in a 4 °C refrigerator and set aside for paraffin sectioning. We set the longitudinal section thickness to 12 μm and detected insoluble carbohydrates (especially polysaccharides and starch granules) with a periodic acid-Schiff (PAS) staining technique. The fertility or sterility of the samples was preliminarily based on the staining results. Images were collected with a light microscope. All experiments were completed by 20 October 2023.

### Identification and analysis of wheat CRK family members

The predicted protein sequences were downloaded from Ensembl Plants to identify CRK genes in wheat (IWGSC RefSeq v1.1: http://plants.ensembl.org/index.html) (*International Wheat Genome Sequencing Consortium , 2018*). The hidden Markov model (HMM) profiles (PF01657; PF00069; PF07714) corresponding to the *CRK* gene family were downloaded from the Pfam database (Pfam 35.0: http://pfam.xfam.org/). The FASTA and GFF3 files containing candidate wheat sequences for the CRK domains were downloaded from the wheat genome (http://plants.ensembl.org). Subsequently, we rebuilt a new wheat-specific HMM file for each HMM file using the HMMER v3.0 suite hmmbuild (*Eddy, 2009*) and used it to identify all the wheat proteins. The identification results of the three hmm files were intersected to obtain a total of 132 CRK proteins with a standard $E$-value $<1\times 10^{-25}$. Then, Expasy Protparma (https://www.expasy.org/) was used to analyze the physicochemical properties of the 132 TaCRKs proteins, including the Compute pI/MW, CDS length, and number of amino acids. The number of transmembrane domains was predicted by the TMHMM server (https://services.healthtech.dtu.dk/services/TMHMM-2.0/) (*Möller, Croning & Apweiler, 2001*). The subcellular localization was predicted by the BUSCA web server (http://busca.biocomp.unibo.it/) (*Savojardo et al., 2018*). A total of 132 candidate CRKs were finally obtained from the wheat genome database using the wheat-specific CRK HMM profiles and further validated by the PFAM database (*Paysan-Lafosse et al., 2023*) and the SMART database (*Letunic, Khedkar & Bork, 2021a*), and these proteins were further analyzed.

### Analyses of phylogeny, gene structure, conserved motifs and protein domain

The CRK protein sequences of tomato (*Solanum lycopersicum*), soybean (*Glycine max*), rice (*Oryza sativa*) and *A. thaliana* were retrieved and downloaded from the

Ensembl Plants database (http://plants.ensembl.org/index.html) and TAIR database (https://www.arabidopsis.org). MEGA-X software (*Kumar et al., 2018*) and the neighbor-joining method (NJ) were used to construct the phylogenetic tree for amino acid sequences with the bootstrap repeated value set to 1,000 times, followed by exporting the newick file and embellishing the phylogenetic tree using iTOL (https://itol.embl.de/login.cgi) (*Letunic & Bork, 2021b*). The gene structure and conserved motif analysis were performed according to previously published methods (*Liu et al., 2021c*). The domain distribution was analyzed using the NCBI-CDD website (https://www.ncbi.nlm.nih.gov/Structure/cdd/wrpsb.cgi). Finally, these files were visualized and optimized using TBtools (*Chen et al., 2020*).

## Analysis of the CRK family promoters, chromosomal location, gene duplication, and synteny in wheat

Gene expression is associated with regulatory elements of promoter sequences. We extracted and obtained the upstream sequences of 95 *CRK* genes CDS of 2-kb length from the genome sequence and used them as identification sites for cis-regulatory elements in the promoter region. These sequences were then submitted to the PlantCARE database (http://bioinformatics.psb.ugent.be/webtools/plantcare/html/) (*Lescot et al., 2002*) for cis-element prediction and visualized with TBtools. Two-way comparison of genomic sequences was performed by the blast tool in the TBtools software. MCScanX Wrapper (*Wang et al., 2012*) was used to identify TaCRKs duplications and the syntenic blocks contacting *CRK* genes of wheat, *A. thaliana*, foxtail millet, rice, and soybean. Subsequently, the gene duplication results and the collinearity comparison map were visualized with TBtools (*Chen et al., 2020*).

## Prediction of putative miRNAs targeting *TaCRK* genes and enrichment analysis

All TaCRK CDSs were uploaded to the psRNATarget website (*Dai, Zhuang & Zhao, 2018*) with parameters set to default, which in turn predicted miRNA targets. Subsequently, the miRNA target network map of *TaCRK* was generated with the assistance of Cytoscape-v3.9.1 (*Shannon et al., 2003*). Gene Ontology enrichment and Kyoto Encyclopedia of Genes and Genomes (KEGG) pathway enrichment of the *TaCRK* genes were analyzed using the EGGNOG-Mapper and TBtools.

## Expression profiles of *TaCRK* genes in different wheat tissues

The wheat RNA-Seq data (IWGSC Annotation v1.1) were downloaded from WheatOmics 1.0 (http://wheatomics.sdau.edu.cn/) to explore the expression profiles of *TaCRK* genes in five different tissues. TBtools software was used to draw the expression heatmap of wheat *CRK* genes.

## Total RNA extraction and quantitative real-time PCR analysis

Total RNA was extracted from wheat anther samples at the mononuclear and trinuclear developmental stages using TRNzol Universal (Tiangen Biotech, Beijing, China) according to the manufacture's requirements. The RevertAid first strand cDNA synthesis kit & DNase I (Thermo Fisher Scientific, Waltham, MA, USA) were used for reverse transcription

according to the manufacturer's instructions. The reverse-transcribed cDNA was used as a template, the wheat actin gene was used as an internal control, and each gene-specific primer was used as the primer. Fluorescence quantification was performed by BIO-RAD CFX Connect™ Fluorescent PCR Detection System according to the Power SYBR® Green PCR Master Mix reaction system. The primer sequence designed with TBtools and Primer Premier 5.0 software are detailed in Table S1. The total reaction volume was 25-µL, which contained 1 µL diluted cDNA and 0.5 µL (10 mM) each primer. The amplification program consisted of an initial step at 95.0 °C for a duration of 3 min. Subsequently, the temperature was set to 95 °C for 10 s, followed by 55 °C for 20 s, 72 °C for 20 s, and 75 °C for 5 s. This cycle was repeated a total of 40 times. The wheat actin gene (AB181991.1) was used as a housekeeping gene for internal control. Three biological replicates of each treatment were performed, and the relative expression of genes was calculated using the $2^{-\Delta\Delta CT}$ method. SPSS version 27.0 (IBM Corp, 2020, Armonk, NY, USA) was used to analyze the significance of differences. $P < 0.05$ indicated significant differences, while $P < 0.01$ indicated highly significant differences.

## RESULTS

### Identification of the *CRK* gene family in wheat

A total of 132 CRK proteins were identified from the wheat genome using Blast comparison and a HMMER 3.0 search with three PFAM files. The intersection was determined and (Fig. 1) analyzed by SMART, removing sequences without CRK structural domains, and combining different transcripts of the same gene (the longest transcript sequence represents the gene) to obtain a total of 95 genes, as shown in Table 1. The genes were named *TaCRK1-1A* to *TaCRK95-U* in sequence according to the order of the gene's position on the chromosome. Analysis of the physicochemical properties of the protein revealed that the molecular weight of wheat CRK protein ranged from 54.35 kD (*TaCRK72-3D*) to 82.75 kD (*TaCRK66-3B*), and the theoretical isoelectric points (pIs) were distributed in the range of 5.21 (*TaCRK44-2D*) to 8.79 (*TaCRK33-2B*). The predicted transmembrane domain (TMDs) analysis of these 132 proteins resulted in 74 proteins containing one TMDs, 51 containing two TMDs, four proteins containing three TMDs, and others without TMDs. There were 117 hydrophilic coefficients less than zero as hydrophilic proteins, and 15 greater than zero as hydrophobic proteins. Furthermore, the predicted subcellular localization indicates that the vast majority of TaCRKs were localized at the plasma membrane, with only a few located in the extracellular space and endomembrane system (Table 1).

### Phylogenetic tree construction, gene structure, motifs, and conserved domains of the *CRK* gene family in wheat

In order to reveal the phylogenetic relationships between wheat and other plant CRKs, an unrooted phylogenic tree of the CRK protein sequences of wheat, *A. thaliana*, rice, soybean, and tomato were constructed. A total of 330 total CRK protein sequences were derived from *A. thaliana* (44), wheat (132), rice (40), soybean (91) and tomato (23). According to the phylogenetic tree branching, the 132 proteins corresponding to the 95 *TaCRK* genes were classified into three major groups, Group 1, Group 2 and Group 3. Group 3 contained

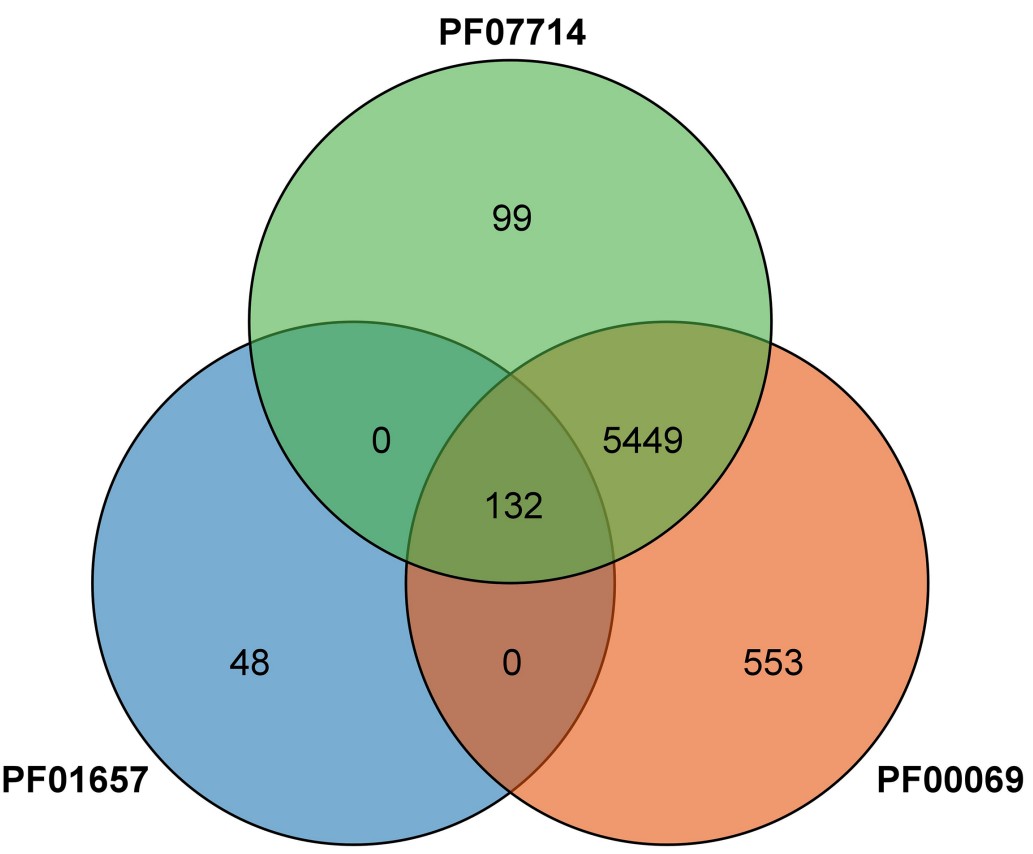

**Figure 1** Venn diagram of the protein identified by the PFAM file involved in the conserved structural domain of the CRK protein.

two subgroups, Group 3-1 and Group 3-2. A total of 14 wheat CRK proteins were present in Group 1 and only two in Group 2; Group 3-1 contained nine CRK proteins and Group 3-2 had 107 wheat CRK proteins. Group 3-2 contained the most wheat CRK and had 29 rice CRK proteins. All of the groups contained rice CRK proteins; for example, Group 1 had four rice proteins, Group 2 had three rice proteins, and subgroup Group3-1 had one rice protein. All of these rice proteins were present on the same branch as the wheat CRK proteins, indicating that the wheat CRK proteins were more closely related to the rice CRK proteins than the other species (Fig. 2).

## Analysis of the gene structure, motifs, and conserved domains of the TaCRK gene family

The amino acid sequences of TaCRK family members were analyzed using the MEME online website to determine their functional regions, and 12 motifs were found. The evolutionary tree of 95 TaCRK family members is divided into three large subgroups, which are indicated by different background colors. Motif 1, 2, 5, 6, 9, 10 are motifs that are present in all protein members. Except for the shared motif, motifs 8 and 11 are found in the first major branch.

**Table 1  Information about the TaCRK genes in wheat.**

| Gene name | Gene Locus | CDS length (bp) | AA[a] | MW[b] (kDa) | pI[c] | TMD[d] | Grand average of hydropathicity[e] | SLP[f] |
|---|---|---|---|---|---|---|---|---|
| TaCRK1_1A | TraesCS1A02G273600.1 | 2,034 | 677 | 72.42 | 6.65 | 1 | −0.064 | plasma membrane |
| **TaCRK2_1A** | TraesCS1A02G422600.1 | 2,028 | 675 | 73.28 | 5.60 | 1 | −0.078 | plasma membrane |
|  | **TraesCS1A02G422600.2** | 1,914 | 637 | 68.4 | 6.14 | 1 | 0.026 | plasma membrane |
| **TaCRK3_1A** | TraesCS1A02G422700.1 | 1,992 | 663 | 72.93 | 7.46 | 2 | −0.108 | plasma membrane |
|  | **TraesCS1A02G422700.2** | 1,995 | 664 | 73.00 | 7.46 | 2 | −0.106 | plasma membrane |
| TaCRK4_1A | TraesCS1A02G422800.1 | 2,073 | 690 | 75.15 | 5.95 | 1 | −0.066 | plasma membrane |
| TaCRK5_1B | TraesCS1B02G283400.1 | 2,037 | 678 | 73.33 | 6.38 | 1 | −0.141 | plasma membrane |
| **TaCRK6_1B** | TraesCS1B02G454000.1 | 1,989 | 662 | 72.85 | 7.09 | 2 | −0.107 | plasma membrane |
|  | **TraesCS1B02G454000.2** | 1,992 | 663 | 72.92 | 7.09 | 2 | −0.105 | plasma membrane |
| TaCRK7_1B | TraesCS1B02G454100.1 | 2,007 | 668 | 72.71 | 7.46 | 2 | −0.135 | plasma membrane |
| TaCRK8_1B | TraesCS1B02G454400.1 | 2,004 | 667 | 72.72 | 5.69 | 2 | −0.056 | plasma membrane |
| TaCRK9_1B | TraesCS1B02G454600.1 | 1,893 | 630 | 68.54 | 7.17 | 1 | −0.204 | plasma membrane |
| **TaCRK10_1D** | TraesCS1D02G273600.1 | 2,046 | 681 | 73.38 | 6.00 | 1 | −0.117 | plasma membrane |
|  | TraesCS1D02G273600.2 | 2,043 | 680 | 73.31 | 6.00 | 1 | −0.12 | plasma membrane |
|  | TraesCS1D02G273600.3 | 1,731 | 576 | 62.29 | 5.69 | 1 | −0.134 | plasma membrane |
|  | TraesCS1D02G273600.4 | 1,545 | 514 | 55.54 | 5.70 | 1 | −0.147 | plasma membrane |
|  | **TraesCS1D02G273600.5** | 1,560 | 519 | 56.04 | 5.83 | 1 | −0.134 | plasma membrane |
| TaCRK11_1D | TraesCS1D02G273700.1 | 2,028 | 675 | 73.46 | 6.17 | 1 | −0.137 | plasma membrane |
| TaCRK12_1D | TraesCS1D02G273800.2 | 1,962 | 653 | 70.05 | 6.26 | 1 | −0.104 | plasma membrane |
| TaCRK13_1D | TraesCS1D02G430800.1 | 1,992 | 663 | 72.01 | 6.07 | 1 | −0.111 | plasma membrane |
| **TaCRK14_1D** | **TraesCS1D02G431000.1** | 1,926 | 641 | 68.84 | 6.28 | 1 | 0.048 | plasma membrane |
|  | TraesCS1D02G431000.2 | 2,019 | 672 | 72.85 | 6.18 | 1 | −0.047 | plasma membrane |
| **TaCRK15_1D** | TraesCS1D02G431400.1 | 2,133 | 710 | 76.79 | 6.00 | 2 | −0.097 | plasma membrane |
|  | **TraesCS1D02G431400.2** | 1,920 | 639 | 68.62 | 6.25 | 1 | 0.055 | plasma membrane |
| **TaCRK16_2A** | **TraesCS2A02G215800.1** | 1,932 | 643 | 70.98 | 6.01 | 1 | −0.141 | extracellular space |
|  | TraesCS2A02G215800.2 | 2,043 | 680 | 75.26 | 6.23 | 2 | −0.1 | plasma membrane |
|  | TraesCS2A02G215800.3 | 2,034 | 677 | 75.03 | 6.35 | 2 | −0.094 | plasma membrane |
| TaCRK17_2A | TraesCS2A02G215900.1 | 2,091 | 696 | 74.66 | 6.18 | 2 | −0.079 | plasma membrane |
| **TaCRK18_2A** | **TraesCS2A02G216300.1** | 2,004 | 667 | 72.72 | 5.95 | 1 | 0.006 | plasma membrane |
|  | TraesCS2A02G216300.2 | 1,698 | 565 | 61.4 | 6.64 | 1 | 0.03 | plasma membrane |
| TaCRK19_2A | TraesCS2A02G216400.1 | 2,001 | 666 | 72.45 | 8.07 | 1 | −0.038 | plasma membrane |
| TaCRK20_2A | TraesCS2A02G216500.1 | 2,040 | 679 | 73.82 | 8.28 | 2 | −0.127 | plasma membrane |
| TaCRK21_2A | TraesCS2A02G216600.1 | 2,007 | 668 | 73.47 | 8.53 | 3 | −0.128 | plasma membrane |
| TaCRK22_2A | TraesCS2A02G216700.1 | 2,058 | 685 | 74.55 | 6.26 | 2 | −0.116 | plasma membrane |
| TaCRK23_2A | TraesCS2A02G216900.1 | 2,010 | 669 | 72.89 | 6.26 | 1 | −0.092 | plasma membrane |
| TaCRK24_2A | TraesCS2A02G217000.1 | 2,064 | 687 | 76.38 | 5.89 | 1 | −0.231 | plasma membrane |
| TaCRK25_2A | TraesCS2A02G217200.1 | 2,100 | 699 | 76.81 | 6.1 | 2 | −0.218 | plasma membrane |
| **TaCRK26_2A** | **TraesCS2A02G217300.1** | 1,998 | 665 | 73.02 | 6.17 | 1 | −0.019 | plasma membrane |
|  | TraesCS2A02G217300.2 | 2,013 | 670 | 73.54 | 6.17 | 1 | −0.014 | plasma membrane |
| TaCRK27_2A | TraesCS2A02G562800.1 | 2,091 | 696 | 76.75 | 5.78 | 3 | −0.202 | plasma membrane |

**Table 1** (*continued*)

| Gene name | Gene Locus | CDS length (bp) | AA[a] | MW[b] (kDa) | pI[c] | TMD[d] | Grand average of hydropathicity[e] | SLP[f] |
|---|---|---|---|---|---|---|---|---|
| | **TraesCS2B02G240800.1** | 1,932 | 643 | 70.84 | 6.00 | 1 | −0.147 | extracellular space |
| TaCRK28_2B | TraesCS2B02G240800.2 | 2,031 | 676 | 74.82 | 6.49 | 2 | −0.092 | plasma membrane |
| | TraesCS2B02G240800.3 | 2,040 | 679 | 75.04 | 6.35 | 2 | −0.098 | plasma membrane |
| **TaCRK29_2B** | **TraesCS2B02G240900.1** | 2,094 | 697 | 75.02 | 5.82 | 2 | −0.053 | plasma membrane |
| | TraesCS2B02G240900.2 | 2,067 | 688 | 74.00 | 5.71 | 2 | −0.067 | plasma membrane |
| TaCRK30_2B | TraesCS2B02G241400.1 | 2,004 | 667 | 72.23 | 6.12 | 1 | 0.058 | plasma membrane |
| TaCRK31_2B | TraesCS2B02G241500.1 | 2,001 | 666 | 72.49 | 7.07 | 2 | −0.061 | plasma membrane |
| TaCRK32_2B | TraesCS2B02G241600.1 | 2,028 | 675 | 72.70 | 5.94 | 1 | 0.015 | plasma membrane |
| TaCRK33_2B | TraesCS2B02G241700.1 | 1,983 | 660 | 70.94 | 8.79 | 1 | −0.003 | plasma membrane |
| TaCRK34_2B | TraesCS2B02G241800.1 | 2,028 | 675 | 73.6 | 6.61 | 2 | −0.138 | plasma membrane |
| TaCRK35_2B | TraesCS2B02G241900.1 | 2,055 | 684 | 75.34 | 8.19 | 3 | −0.096 | plasma membrane |
| TaCRK36_2B | TraesCS2B02G242100.1 | 1,935 | 644 | 70.21 | 6.80 | 1 | −0.118 | plasma membrane |
| | **TraesCS2B02G242100.2** | 2,061 | 686 | 74.56 | 6.40 | 1 | −0.096 | plasma membrane |
| TaCRK37_2B | TraesCS2B02G242300.1 | 2,010 | 669 | 72.81 | 6.05 | 1 | −0.096 | plasma membrane |
| TaCRK38_2B | TraesCS2B02G242500.1 | 2,103 | 700 | 77.07 | 6.24 | 2 | −0.224 | plasma membrane |
| **TaCRK39_2B** | **TraesCS2B02G623700.1** | 2,151 | 716 | 79.38 | 6.06 | 1 | −0.184 | plasma membrane |
| | TraesCS2B02G623700.2 | 2,088 | 695 | 77.05 | 5.83 | 1 | −0.192 | plasma membrane |
| TaCRK40_2B | TraesCS2B02G623900.1 | 1,761 | 586 | 64.64 | 6.06 | 2 | −0.08 | plasma membrane |
| TaCRK41_2B | TraesCS2B02G628100.1 | 1,941 | 646 | 69.05 | 7.46 | 1 | −0.091 | plasma membrane |
| TaCRK42_2B | TraesCS2B02G628300.1 | 1,959 | 652 | 70.85 | 6.82 | 1 | −0.145 | plasma membrane |
| | TraesCS2D02G221400.1 | 2,034 | 677 | 74.91 | 6.68 | 1 | −0.12 | plasma membrane |
| TaCRK43_2D | TraesCS2D02G221400.2 | 1,932 | 643 | 70.87 | 6.26 | 0 | −0.172 | extracellular space |
| | TraesCS2D02G221400.3 | 2,043 | 680 | 75.13 | 6.52 | 1 | −0.126 | plasma membrane |
| | **TraesCS2D02G221400.4** | 1,995 | 664 | 73.38 | 7.49 | 1 | −0.134 | plasma membrane |
| | TraesCS2D02G221500.1 | 1,953 | 650 | 69.54 | 5.21 | 1 | −0.107 | plasma membrane |
| TaCRK44_2D | TraesCS2D02G221500.2 | 1,626 | 541 | 58.41 | 7.85 | 2 | −0.052 | plasma membrane |
| | TraesCS2D02G221500.3 | 2,064 | 687 | 73.82 | 5.78 | 2 | −0.091 | plasma membrane |
| | **TraesCS2D02G221500.4** | 2,070 | 689 | 74.05 | 5.78 | 2 | −0.09 | plasma membrane |
| TaCRK45_2D | TraesCS2D02G221900.1 | 2,004 | 667 | 72.79 | 5.78 | 1 | −0.007 | plasma membrane |
| TaCRK46_2D | TraesCS2D02G222000.1 | 2,007 | 668 | 72.52 | 8.07 | 2 | −0.039 | plasma membrane |
| **TaCRK47_2D** | **TraesCS2D02G222100.1** | 2,109 | 702 | 75.51 | 5.89 | 2 | 0.069 | plasma membrane |
| | TraesCS2D02G222100.2 | 2,112 | 703 | 75.61 | 5.89 | 2 | 0.074 | plasma membrane |
| | **TraesCS2D02G222300.1** | 2,037 | 678 | 74.02 | 6.83 | 2 | −0.165 | plasma membrane |
| TaCRK48_2D | TraesCS2D02G222300.2 | 2,040 | 679 | 74.09 | 6.83 | 2 | −0.162 | plasma membrane |
| | TraesCS2D02G222300.3 | 1,761 | 586 | 64.02 | 7.05 | 2 | −0.108 | plasma membrane |
| | TraesCS2D02G222300.4 | 1,548 | 515 | 56.04 | 8.35 | 2 | −0.15 | plasma membrane |
| TaCRK49_2D | TraesCS2D02G222700.1 | 2,064 | 687 | 74.78 | 6.65 | 2 | −0.125 | plasma membrane |
| TaCRK50_2D | TraesCS2D02G222900.1 | 1,980 | 659 | 71.72 | 5.96 | 1 | −0.037 | plasma membrane |
| TaCRK51_2D | TraesCS2D02G223000.1 | 2,100 | 699 | 76.99 | 6.10 | 3 | −0.221 | plasma membrane |

**Table 1** (*continued*)

| Gene name | Gene Locus | CDS length (bp) | AA[a] | MW[b] (kDa) | pI[c] | TMD[d] | Grand average of hydropathicity[e] | SLP[f] |
|---|---|---|---|---|---|---|---|---|
| | TraesCS2D02G223100.1 | 1,995 | 664 | 72.58 | 6.21 | 1 | 0.008 | plasma membrane |
| **TaCRK52_2D** | TraesCS2D02G223100.2 | 2,010 | 669 | 73.10 | 6.21 | 1 | 0.013 | plasma membrane |
| | **TraesCS2D02G223100.3** | 2,097 | 698 | 76.47 | 6.51 | 1 | −0.007 | plasma membrane |
| TaCRK53_2D | TraesCS2D02G572600.1 | 2,091 | 696 | 76.83 | 6.18 | 1 | −0.217 | plasma membrane |
| TaCRK54_2D | TraesCS2D02G579600.1 | 1,842 | 613 | 67.04 | 7.80 | 2 | −0.1 | anchored component of plasma membrane |
| TaCRK55_3A | TraesCS3A02G021600.1 | 1,914 | 637 | 68.69 | 8.35 | 1 | 0.033 | plasma membrane |
| TaCRK56_3A | TraesCS3A02G493100.1 | 2,079 | 692 | 77.88 | 5.73 | 1 | −0.177 | plasma membrane |
| TaCRK57_3A | TraesCS3A02G493300.1 | 2,052 | 683 | 76.48 | 6.48 | 1 | −0.163 | plasma membrane |
| TaCRK58_3A | TraesCS3A02G493500.1 | 2,136 | 711 | 79.40 | 6.02 | 2 | −0.137 | plasma membrane |
| TaCRK59_3A | TraesCS3A02G493600.1 | 2,097 | 698 | 78.33 | 7.19 | 2 | −0.103 | plasma membrane |
| **TaCRK60_3A** | TraesCS3A02G493700.1 | 2,094 | 697 | 77.76 | 5.5 | 1 | −0.175 | plasma membrane |
| | **TraesCS3A02G493700.3** | 2,148 | 715 | 80.04 | 5.94 | 2 | −0.128 | plasma membrane |
| TaCRK61_3A | TraesCS3A02G493800.1 | 2,139 | 712 | 79.28 | 5.94 | 2 | −0.1 | plasma membrane |
| TaCRK62_3A | TraesCS3A02G493900.1 | 2,115 | 704 | 78.64 | 5.75 | 2 | −0.082 | plasma membrane |
| TaCRK63_3B | TraesCS3B02G024100.1 | 1,923 | 640 | 69.22 | 8.33 | | 0.011 | plasma membrane |
| TaCRK64_3B | TraesCS3B02G024300.1 | 2,049 | 682 | 74.45 | 5.92 | 2 | 0.001 | plasma membrane |
| TaCRK65_3B | TraesCS3B02G555300.1 | 1,902 | 633 | 70.73 | 5.78 | 0 | −0.096 | extracellular space |
| **TaCRK66_3B** | TraesCS3B02G555500.1 | 2,217 | 738 | 82.62 | 5.87 | 1 | −0.111 | endomembrane system |
| | **TraesCS3B02G555500.2** | 2,220 | 739 | 82.75 | 5.87 | 1 | −0.116 | endomembrane system |
| TaCRK67_3B | TraesCS3B02G555600.1 | 2,100 | 699 | 78.4 | 6.45 | 1 | −0.139 | plasma membrane |
| **TaCRK68_3B** | **TraesCS3B02G555700.2** | 2,139 | 712 | 79.66 | 6.09 | 1 | −0.087 | plasma membrane |
| | TraesCS3B02G555700.3 | 2,130 | 709 | 79.26 | 6.09 | 1 | −0.08 | plasma membrane |
| TaCRK69_3B | TraesCS3B02G555800.1 | 2,208 | 735 | 82.26 | 6.60 | 1 | −0.145 | endomembrane system |
| TaCRK70_3D | TraesCS3D02G108500.1 | 2,010 | 669 | 72.21 | 8.21 | 2 | −0.039 | plasma membrane |
| TaCRK71_3D | TraesCS3D02G500800.1 | 2,055 | 684 | 75.97 | 6.35 | 1 | −0.132 | plasma membrane |
| TaCRK72_3D | TraesCS3D02G500900.1 | 1,473 | 490 | 54.35 | 6.28 | 0 | −0.193 | extracellular space |
| TaCRK73_3D | TraesCS3D02G501000.1 | 1,971 | 656 | 73.49 | 6.42 | 1 | −0.065 | plasma membrane |
| **TaCRK74_3D** | TraesCS3D02G501100.1 | 2,142 | 713 | 79.61 | 5.82 | 1 | −0.134 | plasma membrane |
| | **TraesCS3D02G501100.3** | 2,124 | 707 | 78.82 | 5.82 | 1 | −0.139 | plasma membrane |
| TaCRK75_3D | TraesCS3D02G501200.1 | 2,091 | 696 | 78.02 | 7.49 | 2 | −0.099 | plasma membrane |
| TaCRK76_3D | TraesCS3D02G501400.1 | 2,151 | 716 | 79.31 | 5.85 | 2 | −0.111 | plasma membrane |
| TaCRK77_5A | TraesCS5A02G051700.1 | 2,124 | 707 | 75.66 | 6.05 | 2 | −0.028 | plasma membrane |
| TaCRK78_5A | TraesCS5A02G052400.1.cds1 | 2,115 | 704 | 75.37 | 7.83 | 1 | −0.128 | plasma membrane |
| TaCRK79_5A | TraesCS5A02G277200.1 | 1,977 | 658 | 71.70 | 8.63 | 2 | −0.121 | plasma membrane |
| TaCRK80_5A | TraesCS5A02G338500.1 | 1,983 | 660 | 72.22 | 5.44 | 1 | −0.088 | plasma membrane |
| TaCRK81_5B | TraesCS5B02G056000.1 | 2,076 | 691 | 74.66 | 6.72 | 2 | −0.072 | plasma membrane |
| TaCRK82_5B | TraesCS5B02G057700.1 | 2,100 | 699 | 74.69 | 5.89 | 2 | −0.063 | plasma membrane |

Table 1 (*continued*)

| Gene name | Gene Locus | CDS length (bp) | AA[a] | MW[b] (kDa) | pI[c] | TMD[d] | Grand average of hydropathicity[e] | SLP[f] |
|---|---|---|---|---|---|---|---|---|
| TaCRK83_5B | TraesCS5B02G064400.1.cds1 | 2,115 | 704 | 75.32 | 7.14 | 1 | −0.058 | plasma membrane |
| **TaCRK84_5D** | **TraesCS5D02G062800.1** | 2,121 | 706 | 75.59 | 6.11 | 2 | −0.01 | plasma membrane |
|  | TraesCS5D02G062800.2 | 2,124 | 707 | 75.72 | 6.02 | 2 | −0.015 | plasma membrane |
| TaCRK85_5D | TraesCS5D02G063800.1.cds1 | 2,121 | 706 | 75.8 | 8.34 | 1 | −0.132 | anchored component of plasma membrane |
| TaCRK86_6A | TraesCS6A02G294700.1 | 2,004 | 667 | 73.59 | 5.91 | 1 | −0.056 | plasma membrane |
| TaCRK87_6B | TraesCS6B02G417200.1 | 1,953 | 650 | 72.18 | 6.04 | 1 | −0.085 | plasma membrane |
| **TaCRK88_6D** | TraesCS6D02G013400.1 | 2,058 | 685 | 76.36 | 6.69 | 1 | −0.125 | plasma membrane |
|  | **TraesCS6D02G013400.2** | 2,040 | 679 | 75.88 | 6.89 | 1 | −0.119 | plasma membrane |
| **TaCRK89_6D** | **TraesCS6D02G013600.1** | 2,049 | 682 | 74.03 | 6.09 | 2 | −0.094 | plasma membrane |
|  | TraesCS6D02G013600.2 | 2,040 | 679 | 73.76 | 6.25 | 2 | −0.097 | plasma membrane |
| TaCRK90_7A | TraesCS7A02G105100.1 | 1,776 | 591 | 64.41 | 6.04 | 1 | −0.054 | plasma membrane |
| TaCRK91_7B | TraesCS7B02G233100.1.cds1 | 1,962 | 653 | 71.19 | 7.76 | 1 | −0.109 | plasma membrane |
| TaCRK92_7D | TraesCS7D02G099300.1 | 2,001 | 666 | 72.74 | 5.61 | 1 | −0.005 | plasma membrane |
| TaCRK93_7D | TraesCS7D02G328900.1.cds1 | 1,959 | 652 | 70.93 | 7.49 | 1 | −0.104 | plasma membrane |
| TaCRK94_U | TraesCSU02G011500.1 | 1,995 | 664 | 72.97 | 6.33 | 1 | 0.023 | plasma membrane |
| TaCRK95_U | TraesCSU02G066200.1 | 1,974 | 657 | 72.55 | 5.65 | 2 | −0.074 | plasma membrane |

**Notes.**
[a]Length of the amino acid sequence.
[b]Molecular weight of the amino acid sequence.
[c]Isoelectric point of the TaCRK proteins.
[d]Number of transmembrane domains, as predicted by the TMHMM server.
[e]The hydrophilic or hydrophobic nature of the protein; positive values indicate hydrophobic proteins, while negative values indicate hydrophilic proteins.
[f]Protein subcellular localization prediction by the BUSCA web server.
In addition, the bold values indicate selected transcripts representing the corresponding genes.

The second major branch has five TaCRK members, of which *TACRK10-1D* lacks motif 3. In addition, all four proteins lack this motif except for TaCRK5-1B, which contains motif 11. The third major branch has 15 TaCRK members, four of which (TaCRK42-2B, TaCRK55-3A, TaCRK63-3B, TaCRK91-7B) are lacking motif 1 (Fig. 3). The exons and introns of the *TaCRK* gene family were analyzed to further understand the gene structure of this family. There were two to seven exons found in the *TaCRK* gene (Fig. 3). Among these, three genes (*TaCRK36-2B*, *TaCRK4-1A*, *TaCRK69-3B*) contained seven exons, 13 genes contained five exons, 69 genes contained four exons, eight genes contained three exons, and two genes (*TaCRK81-5B*, *TaCRK82-5B*) contained two exons. There were one to nine *TaCRK* genes containing introns. Among them, 64 genes contained six introns and 13 genes contained seven introns. The genes containing nine introns were *TaCRK69-3B* and *TaCRK61-3A*, while those containing only one intron were *TaCRK81-5B* and *TaCRK82-5B*. Most of the genes had upstream and downstream UTR regions, but some genes had missing UTR regions, such as *TaCRK70-3D* and *TaCRK37-2B*. The predictions of the Conserved Domain Database (CCD) showed that most *TaCRK* genes contained the STKc-IRAK conserved domain and the stress-autifung conserved domain.

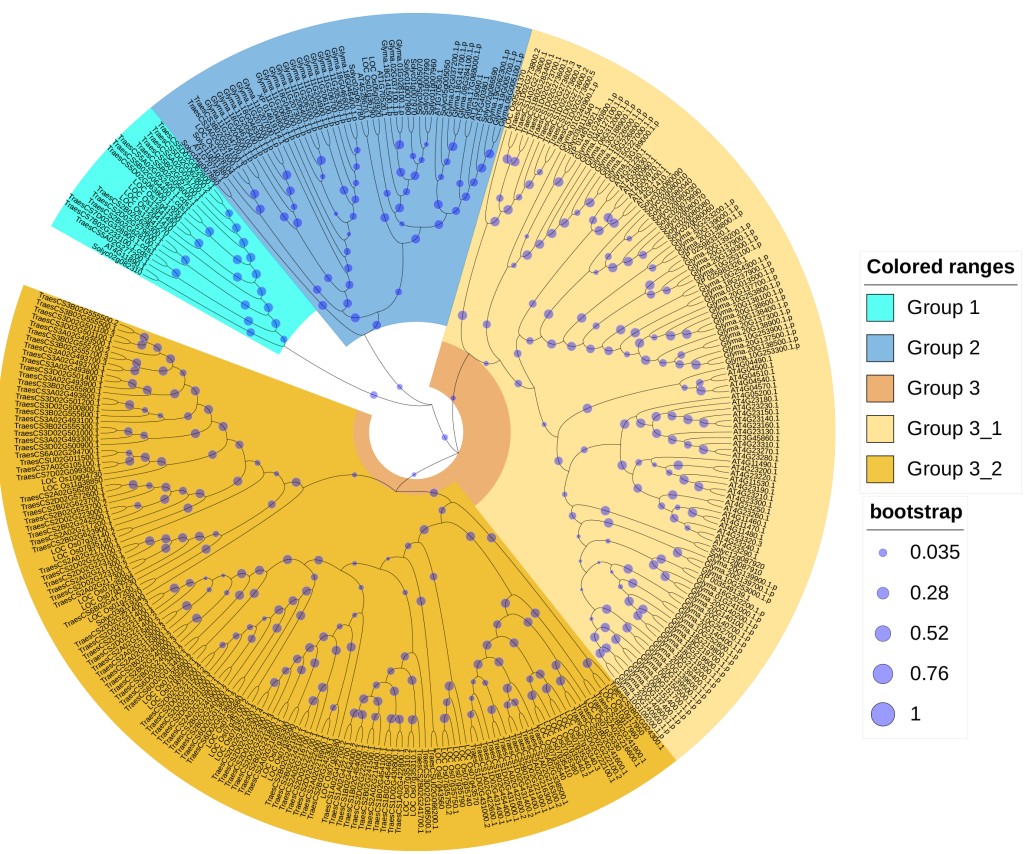

**Figure 2  Phylogenetic tree of CRK gene family in five plant species.** An evolutionary tree was formed by the phylogenetic relationships of 132 predicted TaCRK proteins, 44 *Arabidopsis thaliana* proteins, 40 CRK proteins in rice, 91 CRK proteins in soybean, and 23 in tomato with 1,000 bootstrap replicates by MEGA-X using the Neighbour-Joining method. Different colors are used to mark the subgroups. The new names and accession numbers of wheat CRK proteins are shown in Table 1.

## Distribution of wheat CRK genes on chromosomes and gene duplication

Wheat CRK genes are unevenly distributed on 18 chromosomes, and no *CRK* genes were found on chromosomes 4A, 4B, and 4D. Chromosome 2B contained 15 genes, followed by chromosomes 2A and 2D with 12 genes; chromosomes 6A, 6B, 7A, and 7B were the least numerous, containing only one gene each (Fig. S1). The distribution of TaCRK genes in the same subfamily or evolutionary branches on chromosomes also showed a tendency to be located within the same subgenome. For example, the gene clusters *TaCRK56-3A* to *TaCRK62-3A* are located on the same branch in the phylogenetic tree (Fig. S1; Fig. 3. The Ka/Ks ratios of 109 pairs of *TaCRK* paralogous homologs were calculated to further investigate the effect of evolutionary factors on the TaCRK family. The results showed that the Ka/Ks ratios of all *TaCRK* genes did not exceed 1, with the largest being 0.67 (*TaCRK7-1B*:*TaCRK9-1B*) and less than 0.5 for 99 pairs of genes. This Ka/Ks ratio results suggest that these paralogous homologous genes with ratios less than 1 may have undergone strong purifying selection during evolution (Fig. S2; Table S2). Segmental replication and
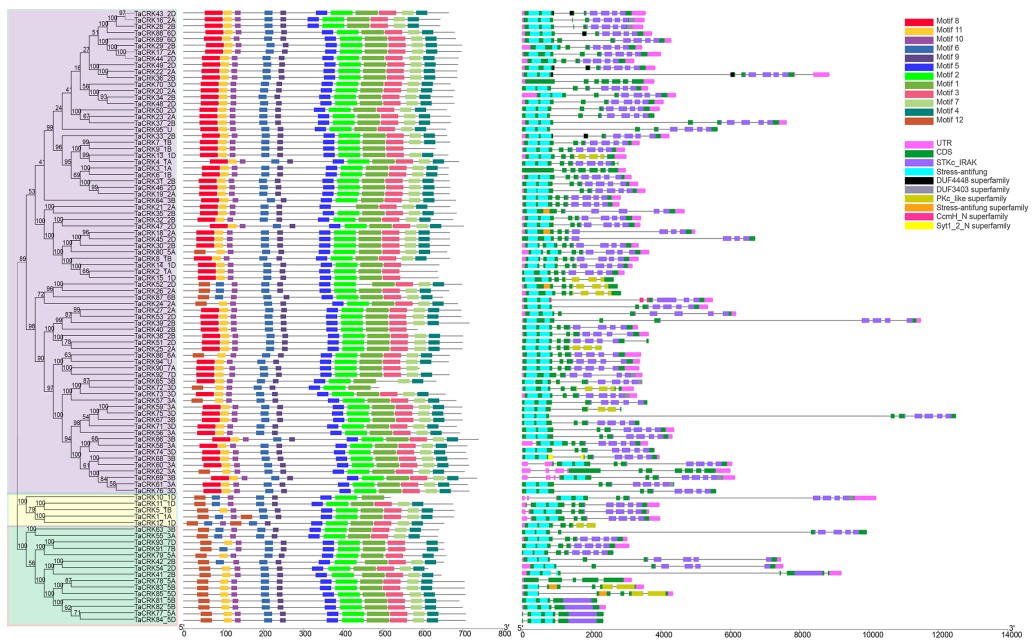

**Figure 3** **Analysis of phylogenetic trees, conserved motifs, gene structures and conserved domains of TaCRK members.** Different colored squares are used to represent the different motifs, CDS, UTR regions and conserved domains in the figure. Black lines represent non-conserved sequences in MEME results and introns in the exon-intron structure, respectively. The phylogenetic tree is constructed in the same method as in Fig. 1.

tandem replication are thought to be the two major reasons for gene family expansion in plants. Seventy-two percent (68/95) of the wheat *TaCRK* members showed duplication events, with 15 tandem duplication events out of 68 duplication events (Fig. 4). For the rest, highly similar genes were found in different chromosomes with segmental duplication events. As shown in Fig. 4, duplication events occurred mainly on chromosomes 2A, 2B, and 2D, while no duplication events were observed on 4A, 4B, and 4D (Fig. 4). In addition, our cluster analysis of homologous genes revealed that almost all homologous genes were in the same evolutionary branch. For example, the segmental repeats *TaCRK1-1A*, *TaCRK5-1B*, and *TaCRK12-1D* have homologous loci in all three partial homologous chromosome groups (A, B, D) and are in the same branch of the evolutionary tree (Fig. 3).

## Collinearity analysis of *CRK* genes between wheat and other representative plants

To investigate the origin and evolution of the *TaCRK* gene in wheat, collinearity analysis of the CRK gene family in wheat and *A. thaliana*, foxtail millet, rice, and *B. distachyon* was performed, and the results showed that wheat and *A. thaliana*, foxtail millet, *B. distachyon*, and rice have three pairs of orthologous genes, 12 pairs of orthologous genes, 15 pairs of orthologous genes, and 11 pairs of orthologous genes, respectively (Fig. 5). In detail, three pairs of orthologous genes (*TaCRK36-2B* and *AT4G23180*; *TaCRK38-2B* and *AT4G23270*; *TaCRK44-2D* and *AT4G23230*) were found in *A. thaliana* and wheat. The others are detailed

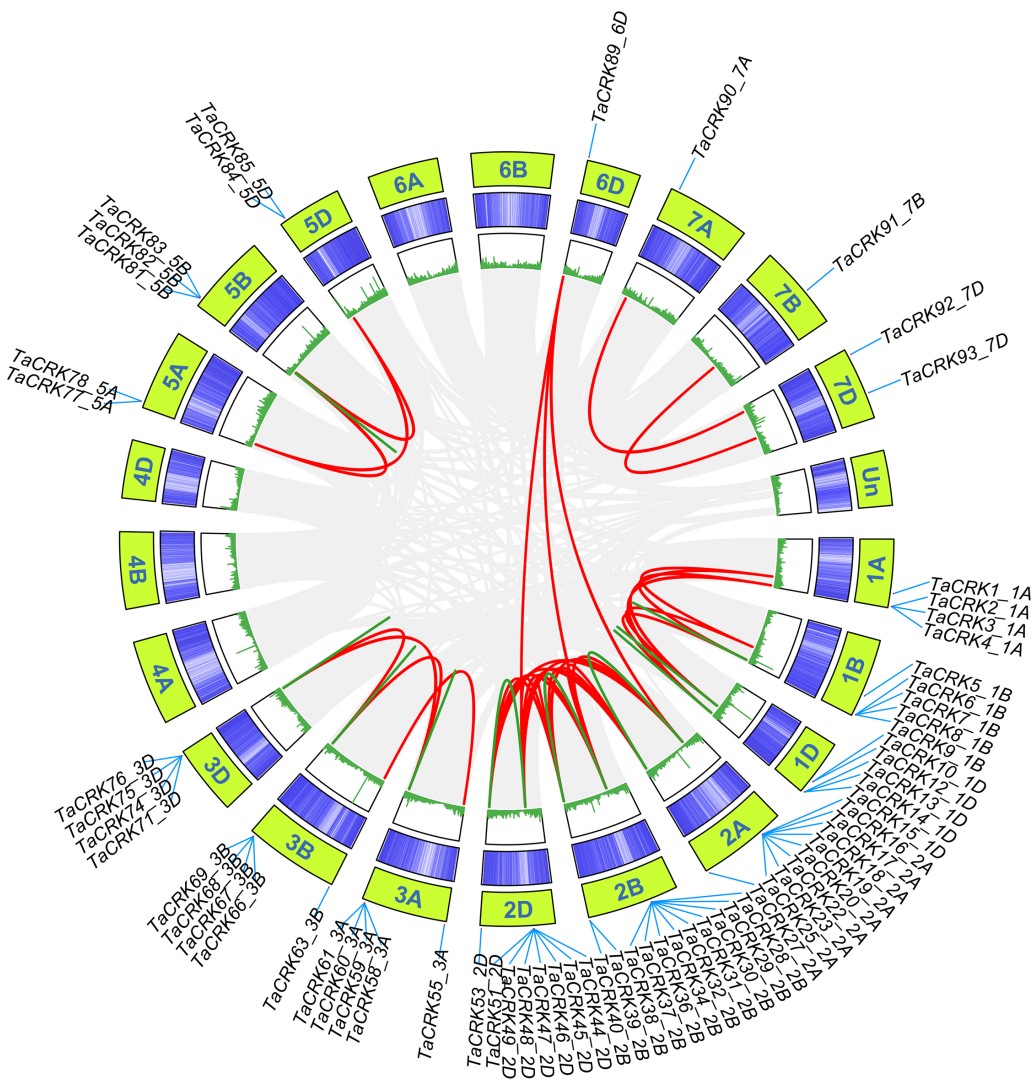

**Figure 4 TaCRK genes family repeat events.** Green, red and gray lines indicate tandem replication, segmental replication and genome-wide background links, respectively. The chromosome number is shown at the bottom of each chromosome.

in Table S3 and Fig. 5. The Ka/Ks values of these *TaCRK* genes were all less than 0.67, indicating that these CRK genes in wheat species have undergone strong purifying selection during evolution and are more functionally conserved. That is, the rate of nonsynonymous substitutions is less than the rate of synonymous substitutions when most nonsynonymous substitutions are deleterious and most nonsynonymous substitutions are purified (Fig. S2).

## Analysis of cis-acting elements of the wheat CRK gene

To further understand the regulatory functions of the genes, the cis-elements in the promoter sequences were analyzed. A 2,000 bp promoter sequence upstream of the CRKs initiation codon was extracted from the wheat genome, and cis-element analysis was performed using the online tool PlantCARE. The TaCRKs promoter region was enriched
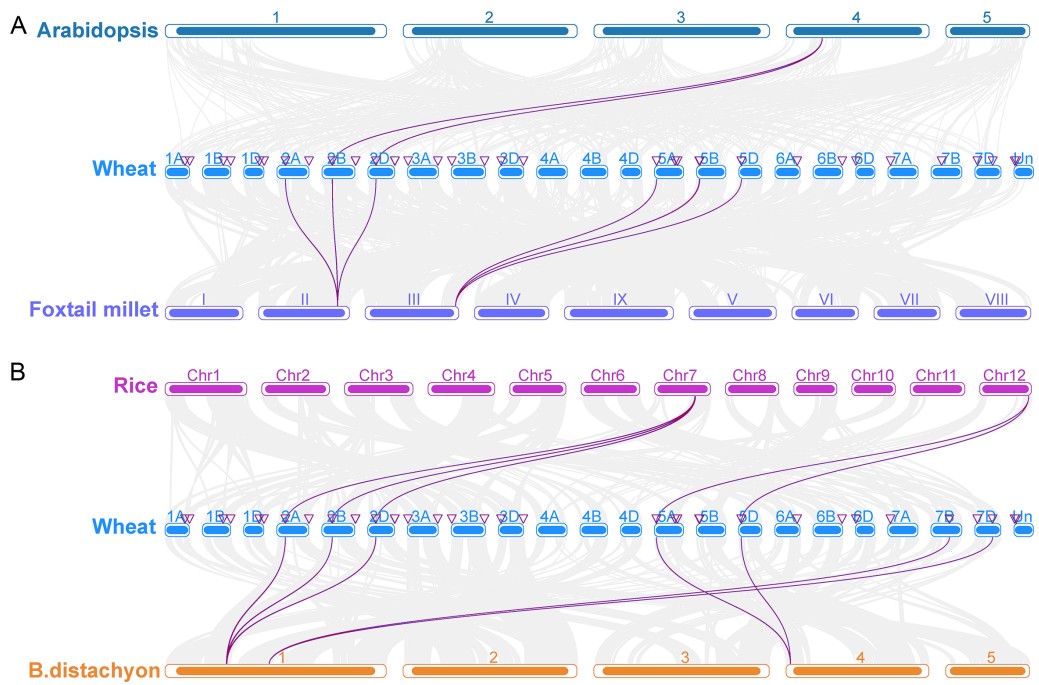

**Figure 5 Collinearity analyses of the TaCRK genes between wheat and four representative plant species (*Arabidopsis*, foxtail millet, rice, and *Brachypodium distachyon*).** Gray lines in the background indicate adjacent blocks in the genomes of wheat and four other representative plants, and red lines highlight gene pairs formed on the genomes of TaCRK genes and four other representative plants.

with cis-acting elements that respond to phytohormone and stress in response to adversity. There are five types of cis-elements related to phytohormone response, namely the abscisic acid response element (ABRE), methyl jasmonate response element (CGTCA-motif; TGACG-motif), salicylic acid response element (TCA), gibberellin response element (GARE-motif, P-box and TCTC-box), and auxin hormone response element (TGA-element). The cis-elements associated with growth and development are the regulatory elements of zein protein metabolism. The cis-elements associated with stress response mainly include anaerobic induction response element (ARE), hypoxia response element (GC-motif), low temperature response element (LTR) and defense and stress response element (TC-rich repeats, MBS). This implies a potential role for the wheat *CRK* gene family in wheat growth and development and in a variety of hormones and stresses (Fig. 6; Table S4).

## GO enrichment and KEGG enrichment analysis of TaCRKs

For the further exploration of the functions of TaCRKs proteins in wheat, we performed GO enrichment and analyzed the cellular composition, molecular functions, and biological pathway categories of these proteins. In the cellular composition, the main enrichment entries were "plasma membrane" and "membrane." In the molecular function category, TaCRKs proteins were highly enriched in "kinase activity" followed by "transferase activity" and "catalitic activity". Analysis of the biological pathways revealed that most TaCRKs

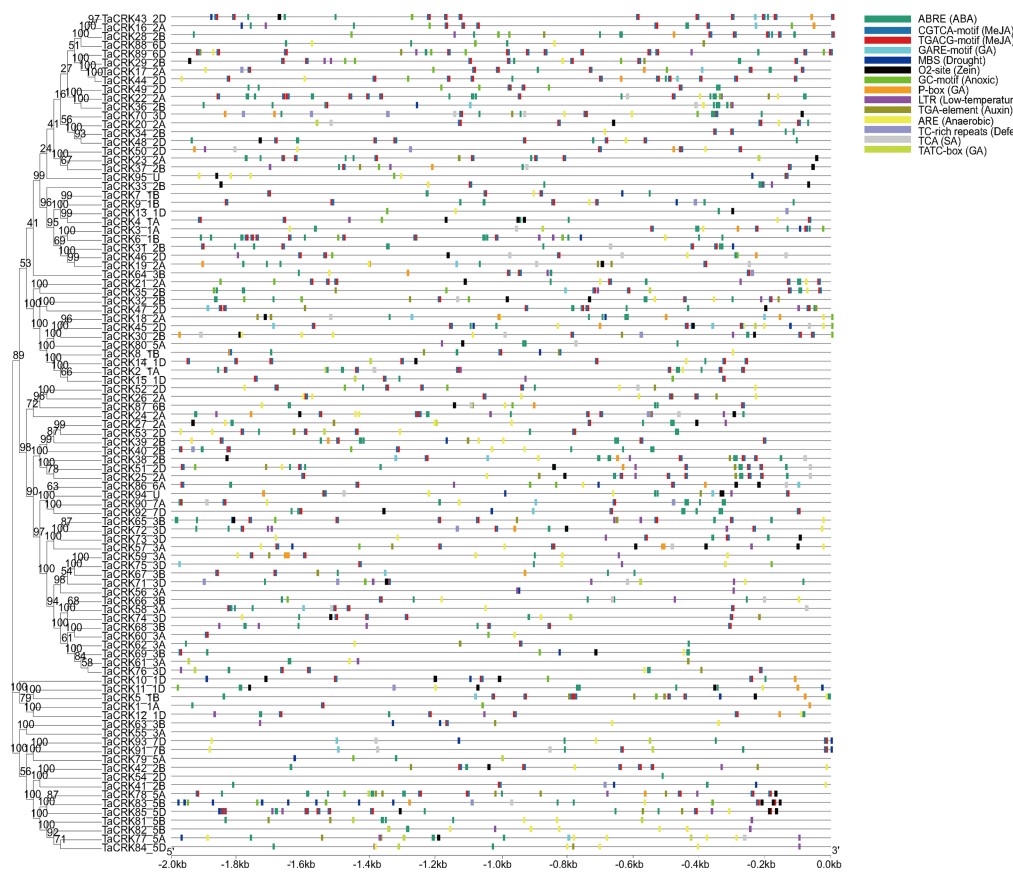

**Figure 6 Cis-acting regulatory elements analysis in TaCRK family members' promoter sequences.**
The promoters of TaCRKs contain cis-acting regulatory elements. 2,000 bp nucleotide length of the gene promoter is indicated on the horizontal axis; color codes indicate different cis-acting elements. On the horizontal axis, the gene promoter length of 2,000 bp is indicated, with different cisacting elements represented by color codes.Phylogenetic tree was constructed by MEGA-X using the Neighbour-Joining method with 1,000 bootstrap replicates.

proteins were mainly assigned to the "cellular protein modification process", "protein metabolic process", and "metabolic process" (Fig. 7A). Subsequently, KEGG enrichment analysis was performed for these TaCRKs proteins." Protein kinases" and "Protein families: metabolism" were the most enriched KEGG enrichment in TaCRK proteins (Fig. 7B).

### The microRNA targets of *TaCRK* genes throughout the genome were identified

MicroRNAs (miRNAs) are a class of small non-coding RNAs that can regulate gene expression at the post-transcriptional level by inhibiting mRNA translation or promoting mRNA degradation. We determined 50 miRNAs targeting 78 genes to better understand how *TaCRK* genes are altered by miRNAs. As shown in the Sankey diagram, one miRNA was (tae-miR9657b-5p) targeted to 15 *TaCRK* genes, which was the largest number, followed by tae-miR164 and tae-miR9780. Similarly, a gene can be regulated by multiple miRNAs, such as *TaCRK68-5B*, *TaCRK94-U*, and *TaCRK52-2D* which target six miRNAs,

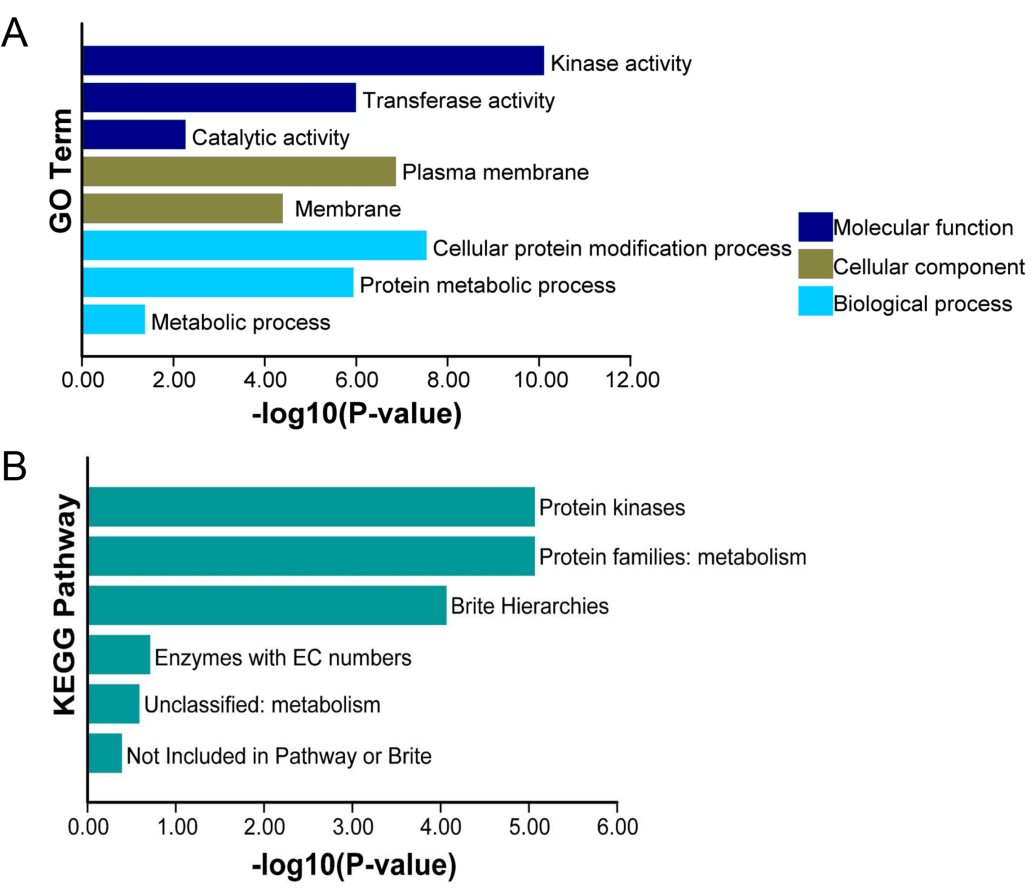

**Figure 7** **This study conducted enrichment analysis of GO and KEGG pathway for the 95 TaCRK genes.** (A) GO enrichment results are presented with green bars, purple bars, and indigo blue bars repre-senting three main categories. (B) KEGG enrichment results are indicated with different color.

five miRNAs, and five miRNAs, respectively (Fig. 8). To improve the visual representation, we mapped the network interactions of genes and miRNAs, and the results showed that five miRNAs were in the hub of the interactions map. These genes include tae-miR9657b-5p, tae-miR9780, tae-miR9676-5p, tae-miR164, and tae-miR531 (Fig. S3A) and further mapped the miRNA targeting sites of *TaCRK85-5D* and *TaCRK52-2D* (Figs. S3B, S3C). Table S5 provides all miRNA targeting sites/genes.

**Expression patterns of wheat *TaCRK* genes in different tissues**

To deeply investigate the expression pattern of *TaCRK* in different tissues, RNA-seq data of Chinese spring wheat varieties were obtained from WheatOmics 1.0. As shown in Fig. 9, *TaCRK* genes were more highly expressed in roots and stem compared with other tissues, with the highest expressed gene also occurring in roots (*TaCRK54-2D*). There are some genes that show high expression in various tissues including root, stem, leaf, spike, and grain, such as *TaCRK36-2B* and *TaCRK49-2D* (Table S6). These two genes are in the same branch of the evolutionary tree (Fig. 3) and are paralogous homologs with a Ka/Ks value of 0.51 (Fig. 4 and Fig. S2). Several other genes were highly expressed in roots, stems,

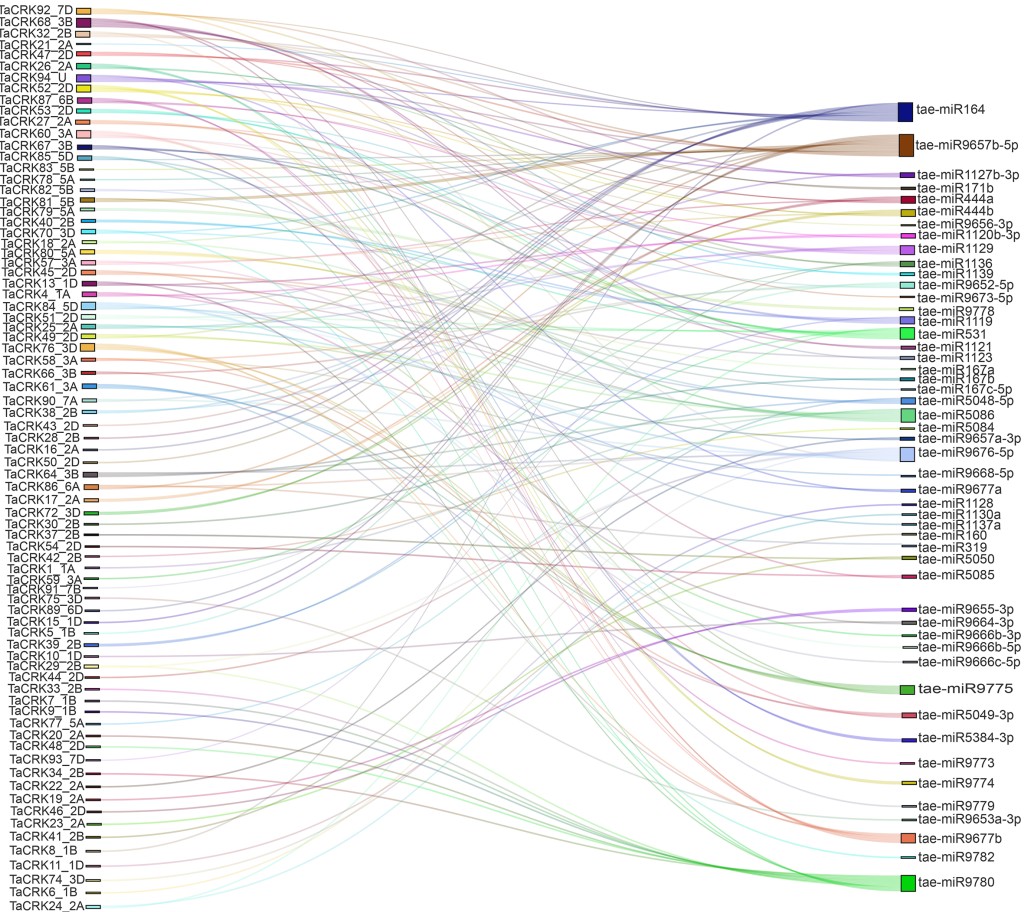

**Figure 8** **Sankey diagram of miRNAs targeting the wheat TaCRK gene.** Different colored modules and connecting lines indicate the relationship of genes and miRNAs that correspond to each other.

spikelets, and grains, but showed low or no expression in leaves, including *TaCRK16-2A*, *TaCRK43-2D* and *TaCRK44-2D*. Figure 9 shows that there is a significant number of genes that exhibit high expression levels in one specific tissue, while displaying lower expression levels in all other tissues or exhibiting low expression levels across all tissues. Notable examples of such genes include *TaCRK61-3A*, *TaCRK29-2B*, *TaCRK79-5A*, and *TaCRK11-1D*.

## The relationship between the wheat CRK family and anther sterility induced by high temperature was investigated using qRT-PCR

High-temperature stress has a negative impact on the growth and reproductive process of wheat plants. During development, HT disrupt the development of stamens, resulting in male sterility. HT-ms plants experience restricted growth, characterized by lower plant height (Figs. 10A, 10B). Additionally, HT stress can also affect the morphology and function of wheat anthers. Normal wheat anthers are slightly larger, and exhibit pronounced dehiscence, facilitating the release and dispersal of pollen. However, under HT conditions, anthers of male-sterile wheat may become smaller and less prone to dehiscence, thereby

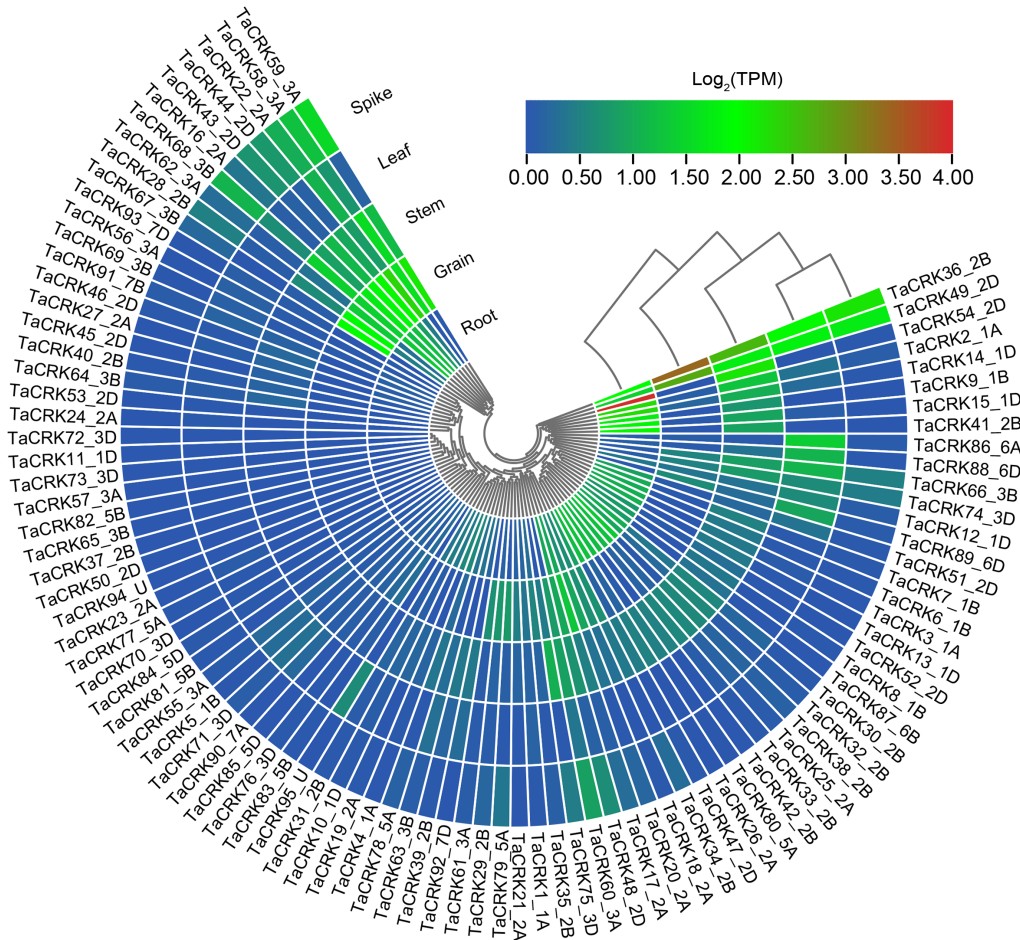

**Figure 9** **Analysis of RNA-seq data reported in WheatOmics was conducted to investigate the differential expression of representative TaCRK genes across different tissues.** The legend indicates the log transcripts per kilobase million (TPM) values. The transcriptome ex-pression results are visualized as a heat map, with colors ranging from blue to green to red. Different colors represent clusters of low and high expression levels.Phylogenetic tree formed using cladistic method that comes with TBtools.

affecting the normal release of pollen (Figs. 10C, 10D). Longitudinal paraffin sections at the monocarpic stage did not exhibit significant differences. However, normal microspores were observed to be slightly larger than sterile microspores, and the epidermis of sterile anthers appeared thicker compared to that of normal anthers (Figs. 10E, 10F). For the trinucleate stage of anthers, normally-developing microspores exhibit abundant starch content in mature pollen grains, while HT-ms anthers show insufficient starch content in microspores, and some even lack starch accumulation. Additionally, there are also structural differences observed in the epidermis between HT-ms anthers and normal anthers (Figs. 10G, 10H).

To further validate the association of these trait changes with *CRK* genes at the gene level, we selected six genes that have collinearity with other species and analyzed their expression changes during the high-temperature male sterility process in anthers. For

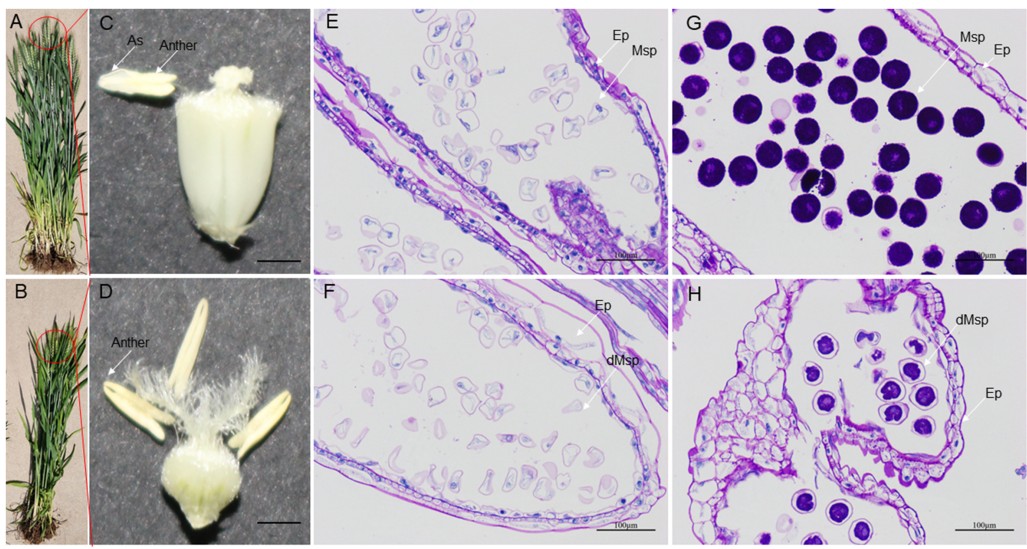

**Figure 10** **Comparison of phenotypic characteristics and anther longitudinal sections between normal and HT-ms Plants.** (A, C) The morphology of normal plants in anthesis on spikelets, anthers and ovary pollination. (B, D) The morphology of HT-ms plants in anthesis on spikelets and anthers and ovary pollination. Longitudinal sections of anther locule of normal and HT-ms anthers at the mononuclear stage (E, F) and the trinuclear stage (G, H). As, anther slit; Ep, epidermis; Msp, microspores; dMSP, degenerated microspore. Bars = five mm in (C, D), and 100 μm in (E–H).

*TaCRK17-2A*, the expression level in the mononuclear stage of sterile anthers was 2.62-FC (fold change) higher than that in Normal anthers, which was significantly higher than that of Normal anthers; whereas the expression in HT-ms anthers at the trinuclear stage was also slightly higher than that of normal anthers but did not reach a significant level (Fig. 11A). The electronic fluorescent pictograph (eFP) demonstrated that this gene was relatively highly expressed in roots (0.92 TPM) and grains (1.42 TPM), while it was relatively low in the spikelets (0.19 TPM) (Fig. 11B). As shown in Figs. 11C–11J, the genes *TaCRK38-2B*, *TaCRK44-2D*, *TaCRK77-5A*, and *TaCRK85-5D* exhibited a significant increase in expression levels at the mononucleate stage of HT-ms anthers, compared to normal anthers, with fold changes of 2.26, 3.40, 4.18, and 3.97, respectively. However, at the trinucleate stage, they showed a decreasing trend but did not reach a significant level. The *TaCRK38-2B* gene exhibits high expression levels in the roots (1.55 TPM) according to the eFP, while its expression in other tissues is generally normalized. *TaCRK44-2D*, *TaCRK77-5A*, and *TaCRK85-5D* all show relatively high expression in grains (3.08 TPM, 0.01 TPM and 0.10TPM, respectively). Additionally, the *TaCRK77-5A* gene also exhibits high expression levels in spikelets and leaves, while the *TaCRK85-5D* gene shows a highly expressed state in roots as well. For the gene *TaCRK93-7D*, the expression level at the mononuclear stage of HT-ms anthers was 7.67-FC higher than normal anthers, showing a highly significant difference. Similarly, at the trinuclear stage, the expression level of HT-ms anthers was slightly higher than that of normal anthers, but the difference did not reach a significant level (Fig. 11K). The eFP results indicate that the gene *TaCRK93-7D*

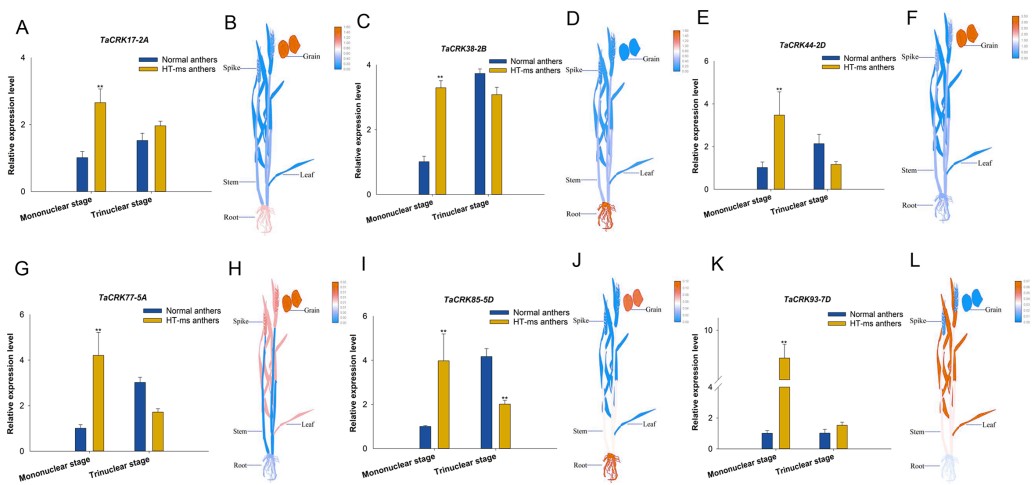

**Figure 11** Differential expression of 6 TaCRK genes in Normal and HT-ms anther tissues was analyzed by qRT-PCR (A, C, E, G, I, and K) and RNA-seq data (B, D, F, H, J, and L). The developmental stages and relative expression levels of normal and HT-ms anthers are shown as $x$- and $y$-axes, respectively. Visualization of electron fluorescence profiles (eFP) of wheat plants was performed with Adobe Illustrator CS5 and TBtools. SPSS Statistics software and SigmaPlot software was used to analyze the data as means of three replicates ±standard error. Capped lines indicate standard error. $*P < 0.05$; $**P < 0.01$.

shows relatively higher expression levels in leaves and stems, while it exhibits relatively lower expression in other tissues (Fig. 11L). Detailed data from the qRT-PCR experiments are shown in Table S7. Overall, all genes at the mononucleate stage of sterile anthers exhibit a highly significant differential high expression compared to Normal anthers. Additionally, compared to normal anthers, there are four genes that show a downregulation trend in expression during the trinucleate stage, while two genes show an upregulation in expression, but the difference in expression levels is not significant compared to normal anthers.

## DISCUSSION

Wheat, a heterozygous polyploid crop, possesses a genome consisting of three sets of subgenomes (A, B, and D) that are highly similar yet distinct. Heteropolyploids like wheat exhibit significant advantages due to their polyploidy nature, but the presence of repetitive sequences and the complex analysis of wheat genes makes it difficult to sequence when compared with other crops (*International Wheat Genome Sequencing Consortium, 2018*). However, in recent years, advancements in genome sequencing technology have enabled the complete sequencing of the wheat genome, leading to the establishment of a comprehensive genome database. This development has facilitated the identification of wheat gene families, exploration of wheat gene function, and discovery of functional genes within the wheat genome (*Xiao et al., 2022*). Cysteine-rich receptor-like kinases possess significant functions in various stress responses, including immunity, oxidative stress, drought tolerance, plant defense response, abscisic acid sensitivity, ultraviolet radiation response and heat stress (*Liu et al., 2021a*). Nonetheless, the understanding regarding the response of CRKs to HT-induced male sterility in wheat was limited. In this research, we
utilized a genome-wide search approach to identify 95 *TaCRK* genes in wheat, surpassing the number found in other species.

The results of the phylogenetic tree analysis indicate that the CRK proteins of wheat can be classified into three branches, with the third branch further divided into two subgroups. Among them, the CRK protein of wheat is present in all three branches, with the second subgroup of the third branch having the highest proportion (Fig. 2). As in other plants such as tomato, and cotton (*Li et al., 2018*; *Liu et al., 2021a*), the *TaCRK* gene is distributed in tandem with in the wheat genome (Fig. 4). It is generally believed that gene duplication, including tandem duplication, whole genome duplication, and segmental duplication, is closely related to the expansion of gene families in flowering plants and has a significant impact (*Freeling, 2009*). The clustering analysis of duplicated genes reveals that almost all duplicate gene pairs are distributed in the same branch, indicating functional similarity between these duplicated genes. The tandemly arranged genes in the *A. thaliana CRK* gene family share similar biological functions, *e.g.*, *AtCRK6* and *AtCRK7* are involved in responding to and transducing extracellular O3 signaling to protect plastids against peroxidative stress (*Idänheimo et al., 2014*). Additionally, gene clusters consisting of the CRK genes are found on chromosomes 1A, 1B, 1D, 2A, 2B, 2D, 3A, 3B, and 3D (Fig. S1). The genes of these gene clusters are in the same major branch in the phylogenetic tree, but the genes within the same gene cluster are not clustered together in the smaller branches (Fig. 3). This indicates that the protein sequences of these genes in the gene clusters may not remain consistent during the process of evolution.

The *CRK* gene family in wheat contains 95 members, 68 of which are arranged in tandem repeats of gene pairs on the corresponding chromosomes with a high degree of sequence similarity. This also suggests that these duplicated genes may have similar biological functions although they may exist on different chromosomes. In higher plants, the genome may undergo gene duplication during chromosome polyploidization leading to tandem duplication events and segmental duplication events, which are considered as the primary drivers for the expansion of gene families in the genome (*Holub, 2001*; *Panchy, Lehti-Shiu & Shiu, 2016*). In the present study, only 15 gene pairs were identified as tandem duplicated genes, while a large proportion of the remaining genes were identified as segmental duplications. This suggests that segmental duplications may play a more important role in the process of gene expansion. Similar phenomena have been observed in other gene families as well. For example, in the WOX gene family of apple, researchers have found that segmental duplications are an important factor contributing to gene expansion (*Xu et al., 2022*). In the tobacco gene family of nonspecific lipid transfer proteins, segmental duplications dominated the evolutionary process (*Yang et al., 2022*). Gene duplication events provide the raw material for the creation of new genes, which in turn provide the basis for the creation of new functions (*Zhu et al., 2014*). Here, most segmental duplications occurred within chromosomal subgroups (Fig. 4), which may be due to the fact that the process of polyploidisation in wheat retains many duplicated chromosomal blocks in its genome. Genes with collinearity within chromosome subgroups have similar expression patterns in different tissues of wheat (*e.g.*, *TaCRK36-2B*:*TaCRK49-2D*; *TaCRK17-2A*:*TaCRK44-2D*, *etc.*; Fig. 9), suggesting that they are functionally similar

or likely to have undergone convergent evolution. Furthermore, the *TaCRK89-6D* gene on chromosome 6D has collinearity with *TaCRK17-2A*, *TaCRK29-2B*, and *TaCRK44_2D* on chromosomes 2A, 2B, and 2D, respectively (Fig. 4). These four genes are located on the same branch of the phylogenetic tree and possess similar motif structure and gene structure (Fig. 3). Interestingly, the expression patterns of the *TaCRK89-6D* gene and three other genes in different tissues of wheat are remarkably different (Fig. 9), suggesting that new genes formed by segmental duplication events outside the chromosomal subgroups may possess distinct functions.

Recent studies have demonstrated the important role of CRK in reactive oxygen species (ROS) production, mitogen-activated protein kinase (MAPK) cascade activation, callose deposition, and programmed cell death (PCD). Plant receptor kinase-like enzymes and ROS contribute to the exchange of information between cells and the external environment. CRK is involved in ROS production and may be part of the ROS sensing mechanism or ROS sensor (*Kimura et al., 2017*). Currently, it is widely believed that there is a close relationship between PCD and the metabolism of ROS. The occurrence of male sterility in plants is associated with the programmed cell death process in tapetum cells. This process, whether accelerated or delayed, can lead to male sterility in plants such as rice, corn, sunflower, wheat, *etc*. ROS act as a signaling molecule that either triggers cell apoptosis or directly induces the process of PCD in cells (*Liu et al., 2018b*; *Zhao et al., 2023*). In the results of our qRT-PCR experiments, the mononuclear stage of high-temperature sterile anthers showed a trend of higher expression of the *TaCRK* gene than that of normal anthers regardless of whether it was highly or lowly expressed in the spikelet (Figs. 11A–11L). However, at the trinuclear stage, four out of six genes (*TaCRK38-2B*, *TaCRK44-2D*, *TaCRK77-5A*, and *TaCRK85-5D*) showed a downregulation trend.

The mitogen-activated protein kinase (MAPK) signaling is intertwined with ROS and abscisic acid (ABA) signaling, which together are involved in contributing to plant adaptation to abiotic stress (*Manna, Rengasamy & Sinha, 2023*). It has been suggested that upregulation of ROS levels in *A. thaliana* cells activates MAPK signaling. In addition, *AtCRK2* may be a negative regulator of the MAPK cascade because MAPK activation is enhanced in *AtCRK2* mutants (*Kimura et al., 2020*). It has also been shown that *AtCRK5* may be involved in ABA signaling in *A. thaliana*, and that the overexpression of *AtCRK5* in early seedlings increased sensitivity to ABA (*Lu et al., 2016*). In this study, the results of the cis-acting elements showed that most of the upstream promoter regions of the CRK genes were characterized by the presence of important elements of ABA response, known as ABRE elements, which implies that the wheat *TaCRK* genes may be involved in the signal transduction of ABA (Fig. 6). This speculation has been validated in *A. thaliana*, where a study demonstrated that *AtCRK45* positively regulates early seedling establishment and response to abiotic stress by regulating ABA biosynthesis (*Zhang et al., 2013*). Furthermore, a transcriptomic study found that the MAPK signaling pathway was involved in the fertility transition of the heat-sensitive inherited male sterile line Zhu1S rice under high temperature conditions (*Chen et al., 2022*). Moreover, an investigation has suggested that the pollen abortion process in wheat K-type CMS may be linked to the regulatory role of the MAPK cascade pathway (*Wu et al., 2023*). These findings suggest that MAPK may play a role in

the process of male sterility, and the co-regulation of CRK, MAPK, and ROS in response to stress also implies that the occurrence of male sterility may be related to CRK. In the current study, the abnormal and especially down-regulated expression of *TaCRK* genes in HT-ms anthers compared with normal anthers, which also suggests that there may be an important association between CRK genes and the HT sterility process in wheat.

Callose is an important cell wall component that exhibits dynamic deposition and degradation during pollen development (*Seale, 2020*). Callose plays an important role in the sexual reproduction in plants, especially in maintaining pollen fertility and survival (*Shi et al., 2015*). In addition, callose is a key regulator of plasma membrane transport in response to stress. It has been reported that *CRK2* enhances plant tolerance to salt stress at the germination stage by promoting callose deposition at the plasma membrane (*Hunter et al., 2019*). Furthermore, callose deposition around the dyads and tetrads was reported to be significantly reduced in ms39 sterile anthers compared to anthers of normal maize plants (*Niu et al., 2023*). However, whether there is a relationship between callose and *CRK* genes as well as male sterility in wheat has not been reported, which is an important direction for our further research in the future. Combining our previous analysis of ROS levels and TUNEL results in high-temperature sterile anthers, we speculate that there may be a correlation between *CRK* gene expression and ROS levels (*Liu et al., 2021d*), and they play a role in the process of high-temperature sterility in wheat anthers. All such presumptions are subject to further experimental confirmation.

## CONCLUSIONS

In conclusion, a total of 95 *TaCRK* genes were identified in the whole wheat genome. The phylogeny tree, gene structure, protein motifs, cis-acting elements, GO and KEGG analyses, collinearity analyses, miRNAs targets of *TaCRK* genes, and expression pattern analyses revealed the conservation and diversity of *TaCRK* genes. Furthermore, we analyzed and discussed the relationship between *TaCRK* genes and HT stress-induced anther sterility in wheat. qRT-PCR results indicated that the expression level of *TaCRK* gene in HT stress-induced wheat mononuclear stage sterile anthers was significantly higher than that in normal anthers. These findings provide some basis for further deeper understanding of the biological roles of individual *TaCRK* genes.

### Funding
This work was supported by the 2019 Postdoctoral Research Project Start-up Funding of Henan Province (No. 226152), the Department of Science and Technology Planning Project of Henan Province (Nos. 222102110412; 202102110173), the 2019 Young Master Teacher Funding Project of Zhoukou Normal University (No. ZKNU20190022). The funders had no role in study design, data collection and analysis, decision to publish, or preparation of the manuscript.

## Grant Disclosures

The following grant information was disclosed by the authors:

2019 Postdoctoral Research Project Start-up Funding of Henan Province: 226152.

Department of Science and Technology Planning Project of Henan Province: 222102110412, 202102110173.

2019 Young Master Teacher Funding Project of Zhoukou Normal University: ZKNU20190022.

## Competing Interests

The authors declare there are no competing interests.

## Author Contributions

- Hongzhan Liu conceived and designed the experiments, performed the experiments, analyzed the data, prepared figures and/or tables, authored or reviewed drafts of the article, and approved the final draft.
- Xiaoyi Li performed the experiments, prepared figures and/or tables, and approved the final draft.
- Zehui Yin performed the experiments, prepared figures and/or tables, and approved the final draft.
- Junmin Hu analyzed the data, prepared figures and/or tables, and approved the final draft.
- Liuyong Xie performed the experiments, analyzed the data, prepared figures and/or tables, and approved the final draft.
- Huanhuan Wu performed the experiments, authored or reviewed drafts of the article, and approved the final draft.
- Shuying Han performed the experiments, authored or reviewed drafts of the article, and approved the final draft.
- Bing Li analyzed the data, authored or reviewed drafts of the article, and approved the final draft.
- Huifang Zhang analyzed the data, authored or reviewed drafts of the article, and approved the final draft.
- Chaoqiong Li analyzed the data, authored or reviewed drafts of the article, and approved the final draft.
- Lili Li analyzed the data, authored or reviewed drafts of the article, and approved the final draft.
- Fuli Zhang analyzed the data, authored or reviewed drafts of the article, and approved the final draft.
- Guangxuan Tan conceived and designed the experiments, prepared figures and/or tables, authored or reviewed drafts of the article, and approved the final draft.

## Data Availability

The raw data are available in the Supplementary Files.

## Supplemental Information

Supplemental information for this article can be found online at http://dx.doi.org/10.7717/peerj.17370#supplemental-information.

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
