# Peer review of "Identification and characterization of the CRK gene family in the wheat genome and analysis of their expression profile in response to high temperature-induced male sterility"

_PeerJ, doi:10.7717/peerj.17370_

## Round 0.1 · original submission · Major Revisions

Dear authors

Revise your manuscript, taking into account the reviewers' suggestions. There are also spelling errors in some places. Correct them as you make revisions.
Regards

**Language Note:** The review process has identified that the English language must be improved. PeerJ can provide language editing services - please contact us at copyediting@peerj.com for pricing (be sure to provide your manuscript number and title). Alternatively, you should make your own arrangements to improve the language quality and provide details in your response letter. – PeerJ Staff

Reviewer 1 ·

Basic reporting

The title is too long. shorten it. Please ensure that the words in the title, keywords, and abstract are not identical
Abstract also is extremely long.
Actually, abstract is a summer of introduction, material and method, results, anc conclusion.
If the introduction had started by discussing the characteristics and significance of the plant under study, it would have been better.
The first passage of the introduction is extremely long; please divide it into at least three passages
include Arabidopsis plant, all genes should be writen in ithalic. for instance, AtCRK4, AtCRK6, AtCRK7, or AtCRK3 and so on...
Why was there a need for such a study? What specific gap in wheat does it aim to address?

Experimental design

Terminated trial date, please write down in material and method.
Please give citation for the SPSS software.
A validation at the protein level is necessary, and the housekeeping gene has not been indicated.
There are too many headings in the materials and methods section. Let's reduce them if possible.
how and where did you do the analysis and obtained those results to create figure 1. (132 CRK proteins were identified from the wheat genome.

Validity of the findings

results and discussions.
132 CRK proteins were identified from the whea.....
in this sentence what is the importance of molecular weight? is it required? Analysis of the physicochemical
properties of the protein revealed that the
210 molecular weight of wheat CRK protein ranged from 54.35 kD (TaCRK72_3D) to 82.75 kD
211 (TaCRK66_3B), and the theoretical isoelectric points (pIs) were distributed in the range of 5.21
212 (TaCRK44_2D) to 8.79 (TaCRK33_2B)
MEGA-X with the neighbor-joining (NJ) method. Citation is required?
This manuscript just focused on CRK protein, not on wheat.
Discussion section is weak, it needs to be developed.
The results section is well-written; however, the discussion section seems somewhat like
an extension of the results. In this section, the differences and similarities of the findings
in this study should be discussed in relation to the literature. The real contribution to science
is in this discussion. While engaging in this discussion, it would be beneficial to explore possibilities,
as there is no certainty in science. It is essential to support your findings with the literature,
and whenever possible, cite the most recent studies.

The conclusion section should directly cover the results and recommendations.
There is no need for excessive explanations. It would be beneficial to check and ensure this."

Reviewer 2 ·

Basic reporting

The manuscript is well-written and, to my extend, meets the journal criteria. Authors have provided all the results sufficiently as well as all data generated (like list of syntenic pairs).

Experimental design

The experimental design and procedures are provided in clear and sequential manner.
The experimental design is common for such type of studies, which is good. Also authors applied a variety of the bioinformatic analyses (more than often used), which is very beneficial for the research.
However, I have specific comments, which are listed below.

Validity of the findings

Author provided all necessary data in the manuscript as well as the source data in the supplementary files. The performed research procedures are clear and correct to my extend. I have specific comments to phylogeny analysis, which are listed below.

As for the Discussion section, authors should integrate their findings more:
- Provide general discussion on gene orthology, duplications etc.
- Specify if the exon-intron structure, protein domain distribution and cis-element distribution was conserved among all the identified CRK. If not, specify whether it was specific to a common phylogenetic group or subgroup of CRKs.
- Discuss the similarities/differences of the identified genes, regarding their common origin (via duplication) and how gene duplicates diverged.

Additional comments

Here I provide point-by-point the question that should be addressed:
L62 – Here and after write A. thaliana, instead of Arabidopsis;
L66 –Starting from here, please, review the usage of italics in gene names;
L118 – cv. Zhoumai 36 instead of Zhoumai 36;
L123 – at -80°C in refrigerator (freezer);
L125 – change ‘4-degree refrigerator’ as mentioned above;
L131 – Specify the genome assembly you have used for genes identification (ID) and add citation, where it was published (if possible);
L142 – Add the citation for TMHMM v2.0;
L130-146 – Carefully review, if the mentioned here databases/tools are not missing the citations to their publications (e.g. BUSCA, TMHMM, etc.). Usually, developers of such tools provide citations for the original publication, where the tool/database was originally presented. Providing only a link is not enough;
L148-150 – Specify the genome assembly ID for the mentioned species;
L151 – cite the respective publication for MegaX software (the program has built-in function allowing you to retrieve the citation);
L153 – The same for iTOL;
L151-153 – Specify the sequence alignment method you used, prior to reconstruct phylogeny, as well as type of sequence you used (gene coding region, translated amino acid sequence). It would be also better to perform phylogeny reconstruction using Maximum Likelihood (ML) method, instead of Neighbor-Joining. In your case, ML would provide more reliable results, as it deals character-wise with the sequences, but not with the general sequence similarity values;
L162 – PlantCARE citation is missing;
L167 – Citation for MCScanX algorithm is missing;
L167-168 – ‘the synteny block of CRK genes’ it is not correct statement. ‘syntenic blocks/regions, contacting CRK genes’ is more correct;
L177 – ID of the sequencing dataset;
L204 – E-value should be mentioned in methods, not here;
L223 – You already mentioned the software in the Methods, don’t duplicate it here;
L226-229 – Name subgroups without low dash (_), use regular dash. Here and after, the same corresponds to gene name also;
L418 – ‘Evolutionary branch’ is not correct formulation
L426-440 – Were the duplicates conserved or they have diverged? Do they share the same gene structure, protein domain distribution or expression patterns? You should disscuss it from the point of the performed synteny analysis. Also, differentiate here duplicates and homeologs (copies, resulting from allopolyploidy).



Figure 2 – Specify the method of phylogeny reconstruction;
Figure 6 – Bootstrap values are overlapping the cladogram (e.g., you represent them in the same way as in Fig. 2). Also, specify the clustering/phylogeny method used in this figure;
Figure 8 – It would be good to avoid overlapping of gene/miRNA names with threads to make the plot more readable;
Figur 9 – Specify clustering method;
Figure 10 – Make scale bar more visible in E, G images;

Reviewer 3 ·

Basic reporting

* The language of the manuscript is clear and easy to follow.
*However, unfortunately, the same attention was not paid to the references made in the text and in the list. A quarter of the publications cited in the text are not listed and vice versa. I’ve marked them with yellow in the attachment.
*Also, the reference list should be written alphabetically.

Experimental design

*The authors did not explain how they established and detected high temperature-induced male sterility.
* What were the concentrations of the RNAs, at least write a range.
*How did they normalize qRT-PCR results? Which gene did they use as an internal control?

Validity of the findings

* A more comprehensive and detailed “Conclusion” section should be written.
* As it is mentioned in the title, an expectation is created regarding the connection between male sterility and the CRK gene family. However, there is a weakness in emphasizing and highlighting the important and strengths of the work.

Additional comments

I commend the authors for their extensive data set and complied laboratory work. To provide more justification of your study discussions must be fluently linked between the overall results and HT-induced male sterility, or else this seems like an additional analysis.

Annotated reviews are not available for download in order to protect the identity of reviewers who chose to remain anonymous.

---

## Round 0.2 · accepted · Accept

You have made the necessary corrections, taking into account the reviewers' and editorial criticisms, and your manuscript is acceptable in its current form. Congratulations

Reviewer 1 ·

Basic reporting

From my side it is ok. The author made all the corrections in entire the manuscript.

Experimental design

From my side it is ok. The author made all the corrections in entire the manuscript.

Validity of the findings

From my side it is ok. The author made all the corrections in entire the manuscript.

Additional comments

From my side it is ok. The author made all the corrections in entire the manuscript.

Reviewer 3 ·

Basic reporting

*The references in the text and the reference list have been updated

Experimental design

*Lack of explanation in the method sections have been completed.

Validity of the findings

*Discussion section is detailed
*Attention was drawn to the importance of the study

Additional comments

*Each comment from the referees was addressed one at a time, and the required changes were put in place.